# R2-P2 rapid-robotic phosphoproteomics enables multidimensional cell signaling studies

Mario Leutert[*] (ID), Ricard A Rodríguez-Mias (ID), Noelle K Fukuda (ID) & Judit Villén[**] (ID)

## Abstract

Recent developments in proteomics have enabled signaling studies where > 10,000 phosphosites can be routinely identified and quantified. Yet, current analyses are limited in throughput, reproducibility, and robustness, hampering experiments that involve multiple perturbations, such as those needed to map kinase–substrate relationships, capture pathway crosstalks, and network inference analysis. To address these challenges, we introduce rapid-robotic phosphoproteomics (R2-P2), an end-to-end automated method that uses magnetic particles to process protein extracts to deliver mass spectrometry-ready phosphopeptides. R2-P2 is rapid, robust, versatile, and high-throughput. To showcase the method, we applied it, in combination with data-independent acquisition mass spectrometry, to study signaling dynamics in the mitogen-activated protein kinase (MAPK) pathway in yeast. Our results reveal broad and specific signaling events along the mating, the high-osmolarity glycerol, and the invasive growth branches of the MAPK pathway, with robust phosphorylation of downstream regulatory proteins and transcription factors. Our method facilitates large-scale signaling studies involving hundreds of perturbations opening the door to systems-level studies aiming to capture signaling complexity.

**Keywords** DIA; MAPK; mass spectrometry; phosphoproteomics; signaling
**Subject Categories** Methods & Resources; Proteomics
**Mol Syst Biol. (2019) 15: e9021**

## Introduction

Cellular signaling is organized as protein networks that respond to changing environments and diverse cellular needs to regulate cellular functions. Protein phosphorylation is integral to the signaling network, targeting most of the components: from sensors to kinases to effector proteins. To obtain a systems-level view of signaling, it is crucial to obtain high-dimensional measurements across the phosphoproteome. High dimensionality, obtained by sampling multiple perturbations and/or many time points after a perturbation, is required to define the edges of the network and obtain mechanistic insight into the structure and dynamics of the network.

Current phosphoproteomic methods allow identifying and quantifying thousands of phosphorylation sites. However, these methods still have limited reproducibility, robustness, and throughput and therefore are not adequate for experiments involving tens to hundreds of perturbations. To facilitate systems biology studies of signaling, we need a robust and automated end-to-end sample preparation workflow, along with mass spectrometry methods that provide systematic sampling at a discovery scale.

Efficient end-to-end phosphoproteomic protocols have recently been developed to simplify, parallelize and, in some cases, automate sample preparation. Humphrey *et al* (2015) developed the EasyPhos platform, a streamlined manual approach that combines protein digestion, desalting, and phosphopeptide enrichment. In the newest version of the protocol sensitivity was improved by performing protein digestion and phosphopeptide enrichment in a 96-well plate (Humphrey *et al*, 2018). Abelin *et al* (2016) automated a workflow for proteomic sample processing and phosphopeptide enrichment over $Fe^{3+}$-IMAC cartridges on a robotic liquid handling platform (Agilent AssayMAP Bravo). These approaches allow deep and robust coverage of the phosphoproteome. However, as most commonly used proteomic sample preparation approaches, these methods rely on sample cleanup by solid-phase extraction on a reversed-phase C18 material. Therefore, they are not compatible with many reagents (chaotropes, detergents, and polymers) commonly used for efficient cell lysis, challenging specimen, or subcellular organelle extractions. Inefficient removal of these reagents from the samples can inhibit enzymatic digestion and/or interfere with LC-MS/MS analysis. Instead, these methods use detergent alternatives, protein precipitation steps, phase-transfer protocols, extensive dilutions, molecular weight cutoff filters, or affinity-based methods leading to trade-offs in flexibility, sensitivity, throughput, and handling (Jiang *et al*, 2004; Manza *et al*, 2005; Kulak *et al*, 2014).

An alternative approach to reversed-phase chromatography for universal proteomic sample preparation is a method called single-pot solid-phase enhanced sample preparation (SP3) (Hughes *et al*, 2014, 2019; Moggridge *et al*, 2018). SP3 uses carboxylate-modified paramagnetic particles that capture proteins via aggregation induced by high-organic solvents (Batth *et al*, 2019). The bead–protein complexes are washed prior to tryptic digestion in the same sample tube. The SP3 protocol is compatible with a wide variety of reagents (detergents, chaotropes, and salts) and allows elution of peptides in buffers that are directly compatible with LC-MS/MS analysis. Due to

Department of Genome Sciences, University of Washington, Seattle, WA, USA
*Corresponding author. Tel: +1 206 543 1880; E-mail: mleutert@uw.edu
**Corresponding author. Tel: +1 206 685 1490; E-mail: jvillen@uw.edu

its high sensitivity, robustness and simple handling process SP3 has found broad application in low input proteomics (Hughes *et al*, 2014; Virant-Klun *et al*, 2016; Sielaff *et al*, 2017; Buczak *et al*, 2018; Cagnetta *et al*, 2018); however, the implementation of SP3 in standard proteomic workflows has been limited, despite its advantages over reversed-phase chromatography methods and its obvious potential for automation and high-throughput sample processing.

Given the high performance and sensitivity of magnetic carboxylated microspheres for manual proteomic sample preparation, we hypothesized that this methodology might be applicable to automated, high-throughput sample preparation using a magnetic particle processing robot. The flexibility of the method should furthermore allow for combination with peptide enrichment methods for analysis of post-translational modifications. Tape *et al* (2014) have shown that it is possible to use a magnetic particle processor in combination with magnetic microspheres to perform fully automated, highly reproducible phosphopeptide enrichment starting from a purified peptide mixture. In the current study, we systematically evaluated experimental parameters to implement an automated, high-throughput sample processing method based on paramagnetic beads that starts from cell lysates, performs protein capture, cleanup, and digestion, and is seamlessly combinable with automated phosphopeptide enrichment. We call our phosphoproteomic sample preparation method R2-P2 (rapid-robotic phosphoproteomics) and the initial proteomics sample preparation R2-P1 (rapid-robotic proteomics).

Reproducibility in phosphoproteomics should be extended beyond sample preparation and into the LC-MS/MS analysis. Most large-scale phosphoproteomics studies so far have employed data-dependent acquisition (DDA) MS measurements. DDA produces extensive data sets; however, its stochastic sampling leaves many missing values when dealing with multiple samples. Data-independent acquisition (DIA) MS is a promising alternative for phosphoproteomics, achieving reproducible sampling, deep phosphoproteome coverage, good quantitative accuracy, and resolution of phosphopeptide positional isomers (Lawrence *et al*, 2016; Searle *et al*, 2019).

We demonstrate the ability of R2-P2 in combination with DIA-MS to accelerate the quantitative analysis of phosphorylation sites in systems-biological studies involving multiple perturbations. Specifically, we study the phosphorylation dynamics of the mating, the high-osmolarity, and the invasive growth mitogen-activated protein kinase (MAPK) pathways, which share many signaling components, but result in very distinct cellular responses. We quantitatively measured the phosphoproteome of *S. cerevisiae* exposed to six different perturbations targeting the MAPK pathway in a three-point time course. We characterized global changes in signaling as well as pathway-specific phosphorylation patterns.

# Results

## An automated magnetic sample preparation method for phosphoproteomics

We aimed at implementing a method for automated, high-throughput sample preparation using carboxylated microspheres on a magnetic particle processing robot that could be seamlessly combined with automated phosphopeptide enrichment on the same robot. For this, we designed the R2-P2 workflow that is conceptually based on the SP3 methodology (Hughes *et al*, 2019), but executable in a 96-well format by a magnetic particle processing robot (KingFisher™ Flex). We configured the platform to perform R2-P1 in a first run and phosphopeptide enrichment for R2-P2 in a second run (Fig 1A). Briefly, the R2-P2 protocol starts by moving the carboxylated beads to a plate that contains protein extracts (e.g., cell lysates) and a high percentage of organic solvent, which promotes protein aggregation and capture on the beads. The carboxylated bead–protein complexes are subsequently passed through three individual plates containing high-organic solvent to wash the bead–protein complexes and remove salts, detergents, lipids, and other contaminants. The clean bead–protein complexes are then moved to a plate containing digestion enzyme in a buffered aqueous solution, and the proteins are digested on the beads at 37°C and simultaneously eluted. We configured the binding and desalting steps to take 30 min and the digestion step 3.5 h. At this point, aliquots for total proteome analysis can be removed, dried, and measured by LC-MS/MS. With the rest of the plate robotic phosphopeptide enrichment using $Fe^{3+}$-IMAC, $Ti^{4+}$-IMAC, $Zr^{4+}$-IMAC, or $TiO_2$ magnetic spheres can be performed. For this, we established a protocol based on previously established methods (Ficarro *et al*, 2009; Tape *et al*, 2014), which takes another 50 min and results in phosphopeptides that can be dried and analyzed by LC-MS/MS. The R2-P1 and R2-P2 detailed protocols and KingFisher™ Flex programs are provided in the Appendix.

## R2-P1 allows efficient protein capture, desalting, and peptide recovery

To establish the R2-P1 automated workflow for protein cleanup and on-bead digestion, we optimized conditions and assessed the efficiency of (i) protein binding to carboxylated beads, (ii) peptide recovery from carboxylated beads, and (iii) digestion efficiency on carboxylated beads using different proteases.

The original SP3 protocol describes binding of proteins to carboxylated beads in 50% acetonitrile (v/v in lysates) under acidic conditions (Hughes *et al*, 2014). But it was later reported that the acidic conditions reduce protein binding (Sielaff *et al*, 2017) and a systematic study of binding conditions recommended ethanol at neutral conditions for optimal binding, recovery, and ease of use (Moggridge *et al*, 2018). To assess the efficiency of protein capture by R2-P1, we adjusted a yeast lysate to 50% or 80% acetonitrile or ethanol, respectively (v/v in lysates), at pH 2 or pH 8. Binding in 50% or 80% ethanol at pH 8 or 80% acetonitrile at pH 2 or pH 8 achieved good protein recovery (Fig EV1A), in agreement with a recent SP3 protocol (Hughes *et al*, 2019). However, diluting lysates to 80% organic solvent limits the total amount of lysate that can be used, and beads clump in high acetonitrile. For these reasons, we chose to use 50% ethanol at pH 8 for protein capture and 80% ethanol for washes in subsequent experiments.

Digestion of proteins on beads in aqueous solution should lead to the elution of peptides from the beads; however, others have observed that this is not always efficient (Batth *et al*, 2019). To evaluate binding and elution efficiency, we tested different elution steps for recovering digested peptides from the carboxylated beads during R2-P1. After digestion in 25 mM ammonium bicarbonate buffer (elution 1), beads were transferred to water and agitated for 5 min

(elution 2) and then transferred to a 2% SDS solution, heated to 95°C, and sonicated (elution 3). The remaining proteins in the lysate, after R2-P1 protein capture, were separately digested (flow through). All fractions were analyzed by LC-MS/MS. The total median peptide intensity in elution 2 was 10% of elution 1, elution 3 was 6% of elution 1, and flow through was 12% of elution 1 (Fig EV1B). The total ion current for the different elution steps confirmed that most of the protein mass is captured by the beads and

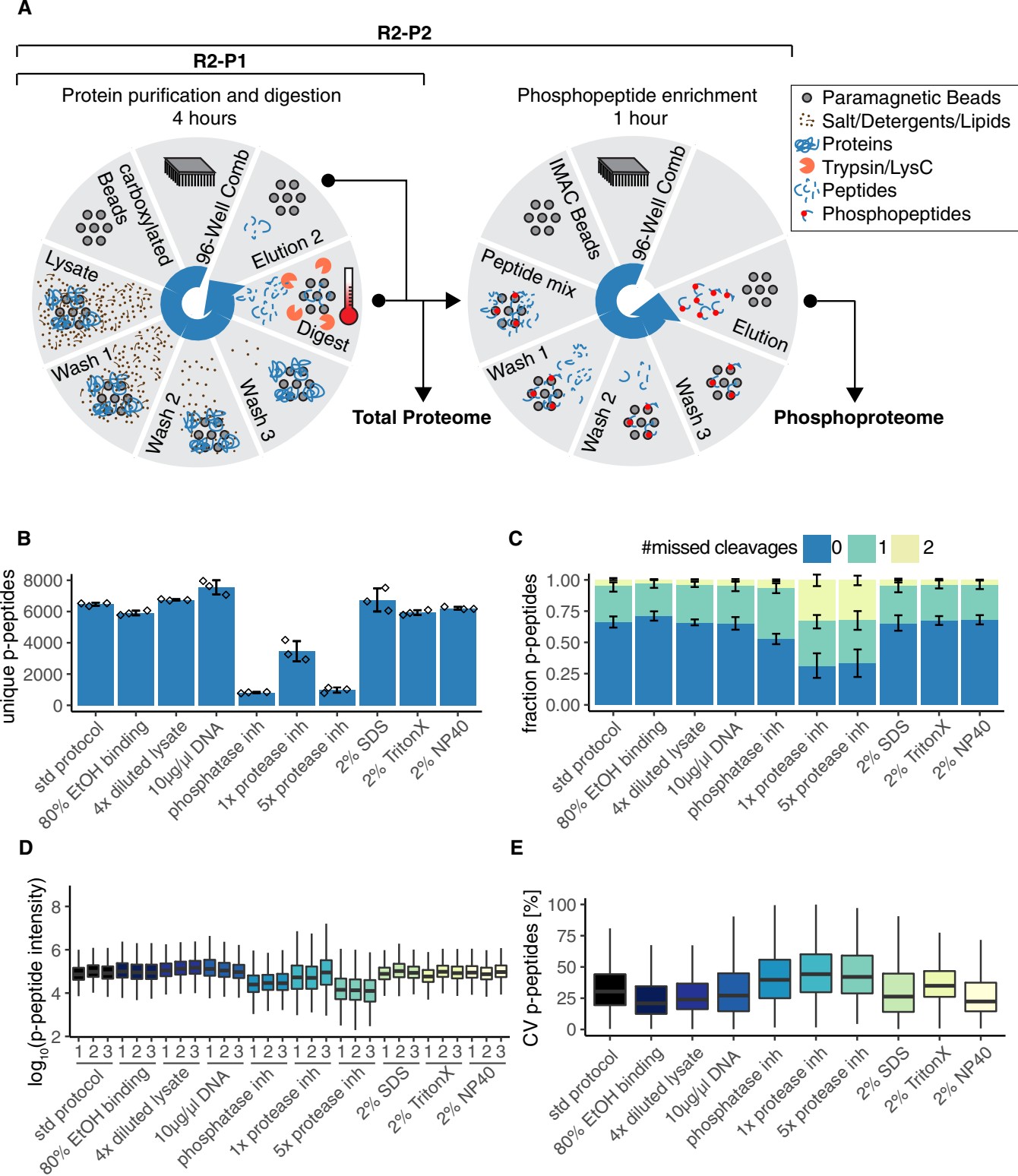

**Figure 1.**

**Figure 1. The R2-P2 method for high-throughput robotic magnetic sample preparation for phosphoproteomic profiling.**

A   KingFisher™ Flex configuration for R2-P2. The robotic configuration allows for loading of eight different 96-well plates. Each plate can be rotated into position under a 96-pin magnetic head that drops down inside the 96-well plate to release, bind, or agitate the magnetic microspheres in solution. In the first robotic run, proteins are captured from lysates by carboxylated magnetic beads, purified, and eluted by digestion at 37°C. Eluted peptides are dried down and can be resuspended for total proteome analysis by LC-MS/MS and/or for automatic phosphopeptide enrichment. Phosphopeptides are enriched using a second robotic run on the KingFisher™ Flex, using $Fe^{3+}$-IMAC, $Ti^{4+}$-IMAC, $Zr^{4+}$-IMAC, or $TiO_2$ magnetic microspheres, and analyzed by LC-MS/MS to obtain the phosphoproteome.

B   200 µg of yeast protein extract in different buffer compositions was processed by R2-P2 followed by DDA-MS peptide analysis. Sample preparation was performed in triplicate. Count of unique phosphopeptides is shown for the different conditions (mean ± SD, $n = 3$).

C   Fraction of phosphopeptides with 0, 1, or 2 missed cleavage sites (mean ± SD, $n = 3$).

D   Boxplot depicting phosphopeptide MS1 signal intensity distributions for each replicate ($n = 3$).

E   Boxplot depicting coefficients of variation (CVs) distributions for phosphopeptide MS1 signal intensities ($n = 3$).

peptides are efficiently eluted during the first elution/digestion step (Fig EV1C). Analysis of the physicochemical properties of the proteins remaining in the flow through of the protein capturing step showed that R2-P1 is slightly biased against small (< 30 kDa) and acidic proteins (Fig EV1D). Based on the proposed mechanism of protein capture by the beads (Batth *et al*, 2019), it is likely that these proteins are not efficiently aggregated on the beads due to their solubility in organic solvents.

Next, we benchmarked different digestion conditions for R2-P1. A total yeast lysate was digested for 3.5 h with either trypsin or LysC at an enzyme to protein substrate ratio ranging from 1:50 to 1:400. Both digestion enzymes provided good and comparable results up to a ratio of 1:200 as judged by the total number of peptides identified (trypsin: $n = 13,198 ± 144$, LysC: $n = 12,868 ± 180$; Fig EV1E) and percentage of fully cleaved peptides (trypsin: 70%, LysC: 86%; Fig EV1F). We set on trypsin at 1:100 as our preferred enzyme for subsequent experiments.

### R2-P2 efficiently enriches phosphopeptides and is robust against a wide range of lysis buffers

We implemented R2-P2 on the same robot by coupling R2-P1 to automated $Fe^{3+}$-IMAC phosphopeptide enrichment (Fig 1A). In a first step, R2-P2 was performed on 200 µg yeast lysate, where proteins were captured either at 50% ethanol (std protocol) or at 80% ethanol (v/v in lysates). In both cases, R2-P2 efficiently enriched phosphopeptides, obtaining slightly more identifications when binding in 50% ethanol ($n = 6,467 ± 104$) versus 80% ethanol ($n = 5,906 ± 146$; Fig 1B).

Next, we systematically evaluated lysis buffer composition compatibility for R2-P2. The following variations on our standard lysis buffer (8 M urea, 150 mM NaCl, 100 mM Tris pH 8) were tested as follows: fourfold dilution of the lysis buffer in water; spike in of 10 µl/µg activated salmon sperm DNA; complementation with phosphatase inhibitors (50 mM sodium fluoride, 10 mM sodium pyrophosphate, 50 mM beta-glycerophosphate, 1 mM sodium orthovanadate); protease inhibitor mix (Pierce); 2% SDS; 2% NP-40; and 2% Triton-X. Samples were analyzed by LC-MS/MS before and after phosphopeptide enrichment. Total proteome analysis showed comparable peptide identifications ($n = 12,000$) and digestion efficiencies for most conditions (Fig EV2A and B). Samples with protease inhibitors were an exception, showing a protease inhibitor concentration-dependent decrease in peptide identification (Fig EV2C) along with increased frequency of missed cleavage sites both with trypsin and with LysC (Fig EV2D). This suggests that certain compounds of the protease inhibitor mix may bind the

carboxylated beads and inhibit enzymatic protein digestion. Of the three detergents tested, only 2% SDS had a slight negative effect on peptide identifications ($n = 9,291 ± 313$), possibly because it was not efficiently removed during the three washes and interfered with LC-MS/MS analysis. If such high SDS concentrations are to be used, an additional carboxylated bead washing step may improve the results.

Starting from 200 µg yeast lysate, R2-P2 showed comparable phosphopeptide identifications for the different lysis buffer compositions ($n = 6,000$) with drastically reduced identifications for the lysis buffer containing phosphatase inhibitors ($n = 814 ± 40$), 5 × protease inhibitor mix ($n = 975 ± 178$) or 1 × protease inhibitor mix ($n = 3,460 ± 644$) (Fig 1B). In line with the results from total proteome analysis, we observed that a high percentage (67–69%) of the phosphopeptides identified contained missed cleavage sites in samples with 5 × and 1 × protease inhibitor mix. This percentage was lower for samples with phosphatase inhibitors (48% missed cleaved) and down to the expected levels in all other conditions (< 35% missed cleaved; Fig 1C). To assess how these conditions perform on a quantitative experiment, we measured MS1 phosphopeptide precursor intensities. All conditions showed comparable values averaging $10^5$ intensity units, except for the samples containing phosphatase or protease inhibitors, which were an order of magnitude lower (Fig 1D). Quantitative reproducibility assessed by the distribution of coefficients of variation (CV) was acceptable for label-free quantification experiments (CV < 30%) and was significantly worse for protease or phosphatase inhibitor containing samples (CV > 40%) (Fig 1E). We conclude that similar to the protease inhibitor binding effect described above, phosphatase inhibitors are also co-enriched by the carboxylated beads. These do not introduce problems for total proteome analysis, but severely impair phosphopeptide enrichment, likely due to competition for $Fe^{3+}$ binding.

Taken together, these results show that the R2-P2 provides a robust method for high-throughput sample preparation for total proteome as well as phosphoproteome analysis that is compatible with a wide variety of detergents and chaotropic agents. Importantly, we identified problems in using protease and/or phosphatase inhibitors in the lysis buffer. *In lieu* of these compounds, we recommend inhibiting endogenous enzymatic activities by using chaotrops in the lysis buffer.

### Benchmarking and scalability of R2-P2

To benchmark our method, we compared its performance to the widely used method of preparing proteomic and phosphoproteomic

samples, which involves in-solution digestion and desalting by solid-phase extraction (SPE) on C18 SepPak cartridges. First, we processed 25 µg yeast protein extract for total proteome analysis by the two methods. Analysis of ~0.5 µg by LC-MS/MS revealed more peptide identifications by R2-P1 ($n = 13,237 \pm 476$) and better overlap between technical replicates (74% peptides identified in at least two out of three replicates) than by SPE ($n = 11,011 \pm 323$ peptides and 63% replicate overlap) (Fig EV3A). While around 50% of all peptides were identified by both methods (Fig EV3B), protein coverage was better with R2-P1 (Fig EV3C). Median MS1 intensity of the identified peptides was slightly higher in the SPE method (median $\log_{10}$ (intensity) = 6.96) compared with R2-P1 (median $\log_{10}$ (intensity) = 6.67; Fig EV3D); however, median CV for the identified proteins was better for R2-P1 (CV = 20.4%) versus SPE (median CV = 24.6%; Fig EV3E). Comparison of molecular weights, GRAVY scores and isoelectric points for the identified peptides (Fig EV3F) and corresponding proteins (Fig EV3G) showed again a slight bias of the magnetic bead method against small acidic proteins.

To assess the scalability of R2-P2 in comparison with SPE coupled with automated IMAC phosphopeptide enrichment, we processed 25, 50, 100, 200, and 400 µg of a yeast lysate in parallel and analyzed 25% of the phosphopeptide-enriched samples by LC-MS/MS. As expected, phosphopeptide intensities increased with input amounts for both methods (Fig 2A). Replicate comparison showed that median CVs for phosphopeptides identified by R2-P2 ranged from 25 to 33% and were comparable to median CVs for SPE (Fig EV4A). To establish the relationship between R2-P2 input protein amount and the resulting phosphopeptide quantifications, we performed linear regressions of MS1 intensities for individual phosphopeptides. Phosphopeptide intensities scaled linearly to the input protein amount in the tested range from 25 to 400 µg (Fig 2B) with median slope of 1.12 (Fig 2C) and median r squared of 0.97 (Fig 2D). This validates R2-P2 for quantitative phosphoproteomics studies. The number of phosphopeptide identifications varied from ~2,900 to ~6,400 (Fig EV4B). Remarkably, R2-P2 achieved more than 4,000 phosphopeptide identifications from 25 µg of starting protein. However, variable enrichment efficiencies precluded phosphopeptide identification comparisons in this experiment (Fig EV4C). For this reason, we performed another R2-P2 experiment for 100, 200, and 400 µg protein input in quadruplicates. In this second experiment, enrichment efficiencies were > 95% for all samples (Fig EV4D), phosphopeptide identifications increased with protein input (Fig EV4E), and MS1 phosphopeptide intensities scaled linearly with protein input as observed before (Fig EV4F).

Taken together, these results show that the performance of R2-P1 and R2-P2 for total proteome and phosphoproteome analysis is comparable with in-solution digestion and C18 SPE purification, the most commonly used proteomic sample preparation method. Phosphopeptide identifications and intensities scaled according to the starting protein amount and were more reproducible with increasing input amounts. We show that R2-P2 is scalable from tens to hundreds of micrograms starting protein.

### Quantitative reproducibility of R2-P1 and R2-P2

To assess quantitative reproducibility of our method within the same robotic batch and over multiple batches, we divided a yeast cell lysate into 25 aliquots of 250 µg and performed five independent batches of R2-P1 and R2-P2, with five replicates each, on five different days. R2-P1 showed consistent identification rates averaging $17,977 \pm 275$ unique peptides (Fig 2E) per sample ($n = 25$ samples). Phosphopeptide identification by R2-P2 was more variable averaging $11,623 \pm 1,097$ unique phosphopeptides ($n = 25$) (Fig 2F). Phosphopeptide enrichment efficiency was consistently high across batches with > 95% of all measured peptides being phosphopeptides (Fig EV5A). Next, as a metric of quantitative reproducibility, we calculated MS1 intensity CVs within the same batch and across different batches. R2-P1 showed high inter- and intra-batch reproducibility for the 7,578 peptides that were quantified in all 25 samples with median CVs of 17–19% (Fig 2G). For R2-P2, 1,021 phosphopeptides were confidently localized and quantified in all 25 samples; inter-batch comparisons had a median CV of 26%, which was comparable to intra-batch comparisons (median CVs 19–27%; Fig 2H). Median Pearson's correlation values for same batch R2-P1 and R2-P2 comparisons were 0.97 and 0.93, respectively, and for different batches were 0.96 and 0.91 (Fig EV5B and C). We note that two out of the 25 R2-P2 samples had markedly lower intensity and lower Pearson's correlations (0.75–0.8) than the other samples, which partially explains the higher variability of R2-P2 compared with R2-P1.

Taken together, R2-P1 shows high reproducibility with no noticeable batch effect. Reproducibility of R2-P2 is slightly lower with regard to phosphopeptide identifications and quantifications, but batch effects were minor.

### Modularity of R2-P2 allows combination with different phosphopeptide enrichment methods

Given the modularity of our robotic approach, we sought to compare and benchmark the performance of different chemistries for phosphopeptide enrichment in combination with R2-P2. For this, we conducted an experiment where triplicates of 250 µg of a yeast protein extract were processed by R2-P1 and phosphopeptides were enriched using either $Fe^{3+}$-IMAC, $Ti^{4+}$-IMAC, $Zr^{4+}$-IMAC, or $TiO_2$ paramagnetic microspheres. All materials performed well, highlighting the versatility of the method. We identified the highest number of phosphopeptides using $Fe^{3+}$-IMAC ($n = 13,417 \pm 368$), followed by $Ti^{4+}$-IMAC ($n = 12,254 \pm 80$), $TiO_2$ ($n = 11,346 \pm 1,122$), and $Zr^{4+}$-IMAC ($n = 10,840 \pm 852$) (Fig 3A). Likewise, $Fe^{3+}$-IMAC achieved the highest enrichment selectivity—on average 94% of all measured peptides were phosphopeptides—(Fig 3B) and recovered the highest fraction of multiple phosphorylated peptides (Fig 3C). Phosphopeptide precursor MS1 intensities were the highest with $Fe^{3+}$-IMAC and $TiO_2$ (Fig 3D), and all enrichments showed good reproducibility with MS1 intensity median CVs around 25% (Fig 3E). Overlap of all identified phosphopeptides for the different enrichment methods showed that 51% of all phosphopeptides were exclusively identified by a single enrichment type (Fig 3F), which was only marginally (10%) higher than when overlapping multiple replicates of all enrichment types combined, reflecting stochastic MS sampling. These results demonstrate the modularity and versatility of R2-P2 with regard to different methods and materials for phosphopeptide enrichment.

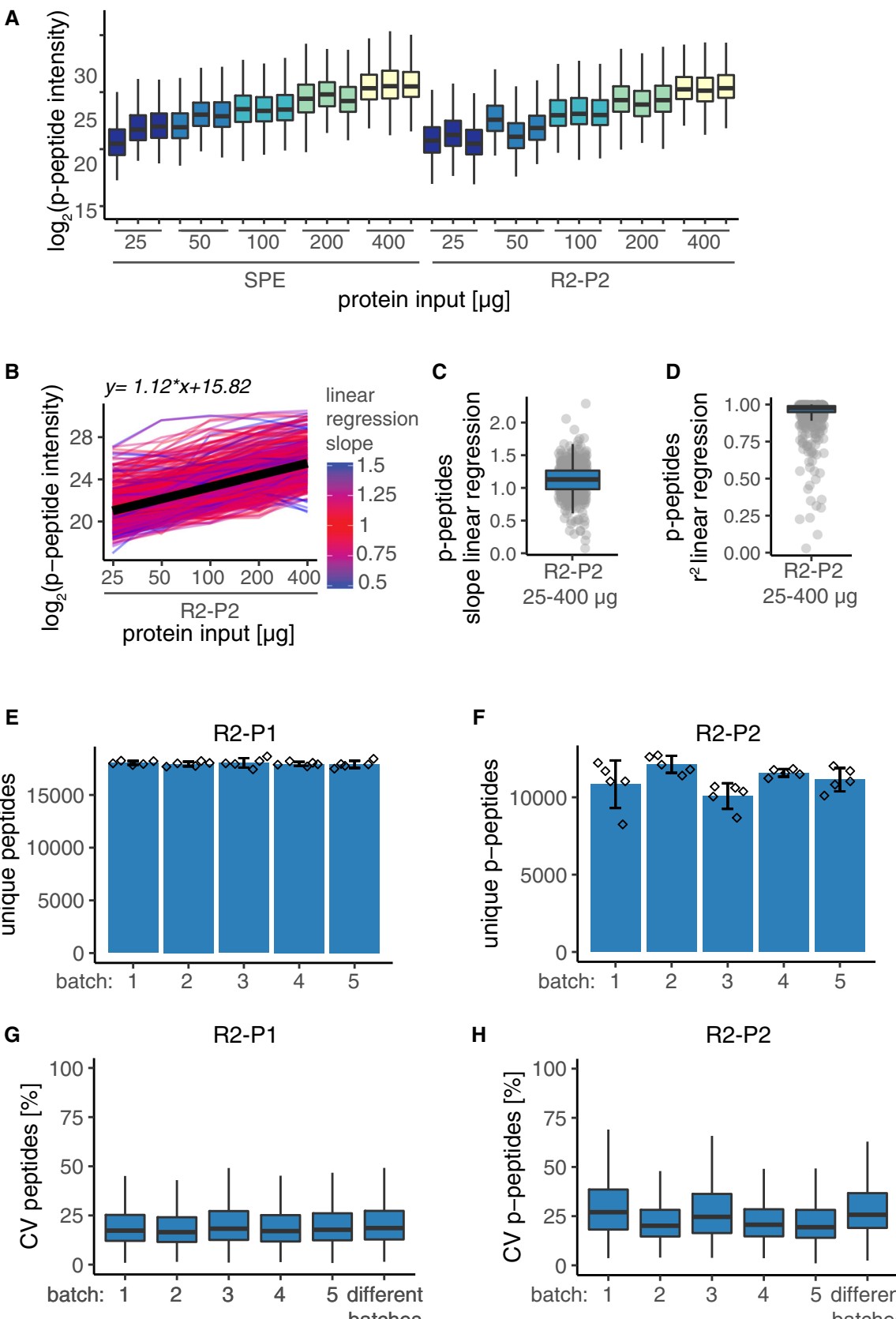

**Figure 2.**

◀

**Figure 2. R2-P2 allows scalable and reproducible phosphoproteomics.**

- A Different amounts (25, 50, 100, 200, and 400 μg) of yeast protein extract were processed by R2-P2 or by in-solution digestion and reversed-phase C18 SPE desalting, phospho-enriched by robotic $Fe^{3+}$-IMAC followed by peptide analysis with DDA-MS. Sample preparation was performed in triplicate. Boxplot depicting MS1 signal intensity distributions for phosphopeptides detected in all samples.
- B Line plot of $\log_2$ phosphopeptide intensity versus protein input amounts with each line representing an individual phosphopeptide (mean, $n = 3$). For every phosphopeptide, a linear regression was performed and lines are colored according to the slope of the linear regression ($n = 553$). Black line represents the linear regression of the median phosphopeptide intensities, and the corresponding function is shown on top.
- C Boxplot depicting the distribution of slope values resulting from linear regressions of individual phosphopeptides shown in (B) ($n = 553$).
- D Boxplot depicting the distribution of $r$ squared values resulting from linear regressions of individual phosphopeptides shown in (B) ($n = 553$).
- E Number of unique peptides identified for R2-P1 performed on five different days (mean $\pm$ SD, $n = 5$).
- F Number of unique phosphopeptides identified for R2-P2 performed on five different days (mean $\pm$ SD, $n = 5$).
- G Boxplot depicting CVs of peptide MS1 signal intensities for reproducibility analysis of R2-P1 within the same batch (mean $\pm$ SD, $n = 5$) and between different batches conducted on different days ($n = 5$). For each peptide, one replicate per batch was randomly chosen for inter-batch comparisons.
- H Boxplot depicting CVs of phosphopeptide MS1 signal intensities for reproducibility analysis of R2-P2 within the same batch ($n = 5$) and between different batches conducted on different days ($n = 5$). For each peptide, one replicate per batch was randomly chosen for inter-batch comparisons.

## Application of R2-P2 to a multiple-perturbation experiment targeting the MAPK pathway

To showcase an application of R2-P2, we combined R2-P2 with DIA-MS to systematically study cellular signaling along the MAPK axis, including the pheromone response/mating pathway, the high-osmolarity stress response pathway, and the invasive growth response pathway.

Experiments were performed in the Σ1278b *Saccharomyces cerevisiae* strain. Σ1278b has a functional invasive response pathway that can be induced by nutrient limitations and certain alcohols, whereas most laboratory yeast strains have acquired mutations that compromise the invasive growth response (Cullen & Sprague, 2012). Yeast cultures were exposed to one of three distinct stimuli (alpha factor, sodium chloride, and 1-butanol) or three media replacements (replacement of glucose with galactose, glucose limitation, and nitrogen limitation) or left untreated, for 10, 30, and 90 min, in three biological replicates. Alpha factor induces the MAPK mating pathway, and NaCl induces the MAPK high-osmolarity glycerol (HOG) pathway. Replacement of glucose with galactose, glucose and nitrogen limitation, and 1-butanol have been described to activate the invasive growth pathway via MAPK and/or three other pathways (RAS/PKA, SNF, and TOR; Cullen & Sprague, 2012). For every sample, 400 μg yeast protein extract was processed using R2-P2 with both $Fe^{3+}$-IMAC and $Ti^{4+}$-IMAC phosphopeptide enrichments (Fig 4A).

For the LC-MS/MS measurement, we first acquired a limited number of DDA measurements to build a spectral library. For this, we pooled three biological replicates and the three time points for each condition and separately measured the $Fe^{3+}$-IMAC and $Ti^{4+}$-IMAC fractions. These 12 injections resulted in the measurement of 31,853 unique phosphopeptides (Fig EV6A, Dataset EV1). Additionally, seven injections of a total IMAC sample pool were dedicated to gas-phase fractionation by staggered narrow-window DIA and were later used to generate a chromatogram library that catalogs retention time, precursor mass, peptide fragmentation patterns, and known interferences that identify each peptide within a specific sample matrix (Fig 4A). Subsequently, a single injection of the $Fe^{3+}$-IMAC fraction for every condition ($n = 63$) was measured by wide-window DIA for the label-free quantification experiment. The MS measurements resulted in 82 injections with 60-min peptide separation gradients; together with the sample preparation (starting from

cell pellets), the experiment was completed in 8 days (Fig 4A). For the analysis of the acquired data, we used Skyline to generate the spectral library from the DDA runs (MacLean *et al*, 2010), EncylopeDIA to generate the chromatogram library from the narrow-window DIA runs (Searle *et al*, 2018), and Thesaurus to identify, site-localize, and quantify phosphopeptides in the wide-window DIA runs (Searle *et al*, 2019).

An average of 7,880 phosphopeptides was quantified for each condition, and ~40% of all measured phosphopeptides had all sites confidently localized (Fig 4B, Dataset EV2). Collectively, we quantified 8,314 phosphopeptide isoforms (peptides containing the same combination of phosphosites) across all conditions (Dataset EV3). Median phosphopeptide quantification CVs for the biological triplicates were between 13 and 33% (Fig EV6B). Correlations of biological replicates (median Pearson's correlation > 0.9) were higher than for different time points and treatments (median Pearson's correlation 0.7–0.8), allowing us to identify true biological changes (Figs 4C and EV6C). Intriguingly, biological variability was higher for the 90-min time point across most perturbations (Figs 4C and EV6C). Untreated, alpha factor, NaCl-, and BuOH-treated samples were well correlated, whereas the nutrient limitation conditions showed more differences, judged by Pearson's correlation (Fig EV6C). Looking at phosphopeptide isoforms that were significantly changing in abundance compared with the untreated (two sample *t*-test, permutation-based FDR < 0.01, fold change > 1.5) revealed that nutrient changes (i.e., low nitrogen, low glucose, and replacing glucose with galactose) had the most profound impact on the phosphoproteome (Fig 4D).

### Functional characteristics of regulated phosphorylation events

To assess the similarity between the seven conditions, we conducted PCA analyses for each time point. This analysis showed good separation between different stimuli and close similarity between biological replicates for the 10- and 30-min time points (Fig 5A). The biological replicates for the 90-min time point were more spread, particularly for the osmotic stress with NaCl, suggesting that they may follow different stress recovery trajectories (Fig 5A).

Next, we performed time-resolved hierarchical clustering of significantly regulated phosphopeptide isoforms (Fig 5B). Generally, clusters for the 10- and 30-min time points were selective to one

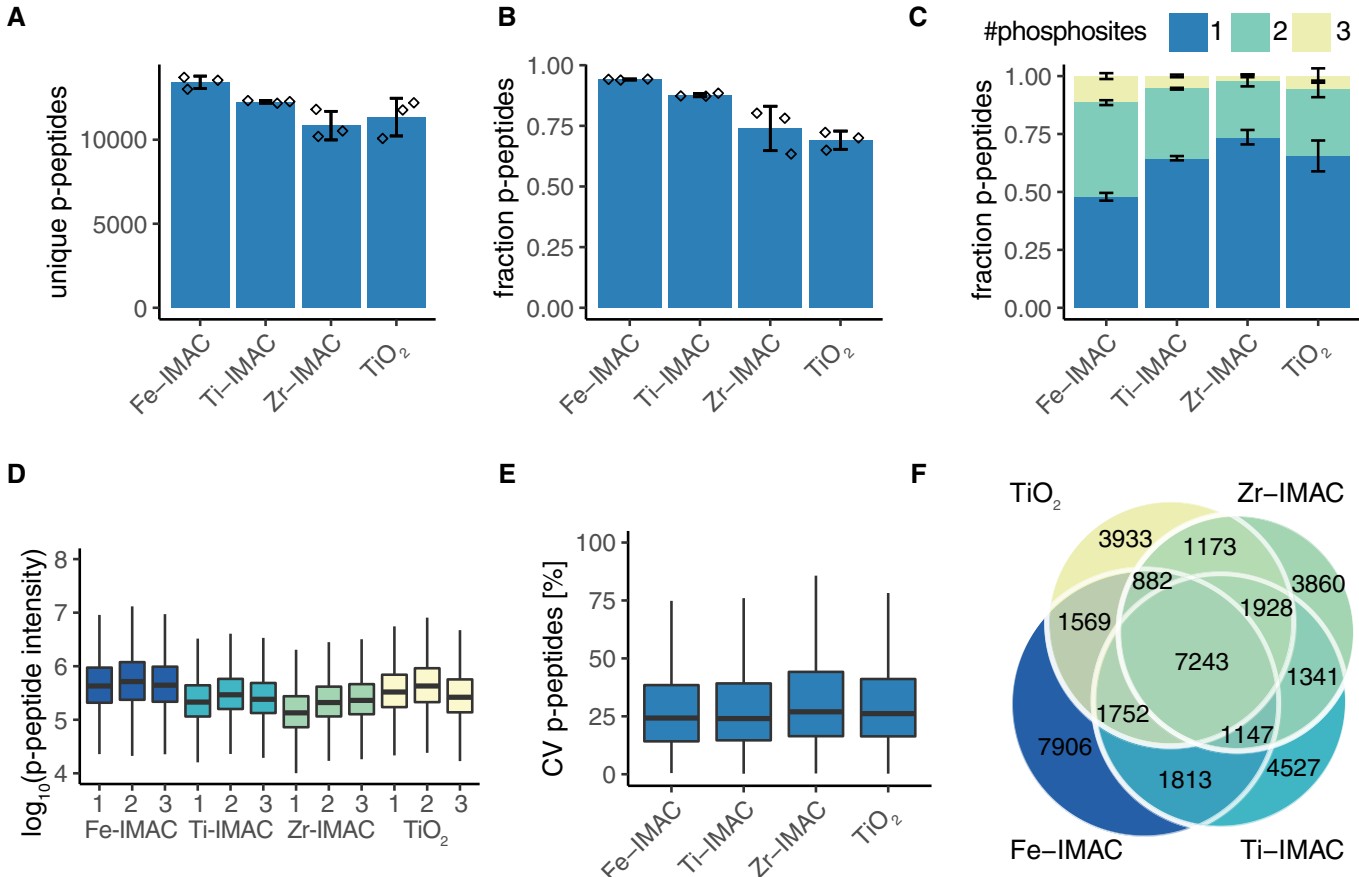

**Figure 3.** Combination of R2-P2 with different phosphopeptide enrichment methods allows for deep phosphoproteomic profiling.

250 µg of yeast protein extract was processed by R2-P2 using either $Fe^{3+}$-IMAC, $Ti^{4+}$-IMAC, $Zr^{4+}$-IMAC, or $TiO_2$ magnetic beads and followed by peptide analysis with DDA-MS. Phosphopeptide enrichment was performed in triplicate.

A  Number of unique phosphopeptides identified by the different enrichments (mean ± SD, $n = 3$).
B  Phosphopeptide enrichment efficiency shown as the fraction of phosphorylated peptides over total peptides (mean ± SD, $n = 3$).
C  Fraction of phosphopeptides with 1, 2, or 3 phosphorylation sites (mean ± SD, $n = 3$).
D  Boxplot depicting distributions of phosphopeptide MS1 signals ($n = 3$).
E  Boxplot depicting distributions of phosphopeptide MS1 signal CVs ($n = 3$).
F  Venn diagram of identified phosphopeptides by the different phosphopeptide enrichment methods.

condition, contrasting the 90-min clusters composed of phosphosites regulated in multiple conditions. We observed partially overlapping response for the nutrient limitation conditions, with low glucose and low nitrogen overlapping at 30 min (clusters 30-2), and low glucose and galactose overlapping at 90 min (clusters 90-3). Furthermore, all nutrient switch conditions showed fast downregulation of a common set of phosphosites (clusters 10-5). NaCl treatment had the most pronounced treatment-specific effect at 10 min (clusters 10-4), and the effect of alpha factor treatment was most pronounced at 10 and 30 min (clusters 10-8, 30-7).

To learn about the biological processes, cellular components, and pathways modulated by the different treatments, we performed a GO enrichment analysis of the regulated phosphopeptide isoforms in every condition compared with untreated (Fig 5C). MAPK signaling pathway is enriched for most treatments, with alpha factor showing the strongest and most sustained enrichment. All treatments showed cellular component enrichment of the cytoskeleton

and the site of polarized growth for most time points. For alpha factor stimulation, we observed enrichment in proteins involved in cell cycle, cell budding, cell division, and establishment of cell polarity at the 90-min time point, capturing the functional effectors of the mating response, which involve cell cycle arrest, changes in cellular architecture, and polarized growth. The osmotic stress condition showed modest enrichment of stress-responsive proteins and proteins associated with establishment of cell polarity at the early time point and no defined GO signature for later time points, possibly reflecting its fast and transient signaling response. Butanol, galactose, low glucose, and low nitrogen all showed extensive regulated phosphorylation events for cell division, cell budding, cell growth, transport, and endocytosis, with enriched localization at the plasma membrane. In butanol and the nutrient limitation conditions, we observe enrichment of phosphoproteins associated with filamentous/pseudohyphal growth terms (e.g., invasive growth), which followed similar temporal profile as the induction of the

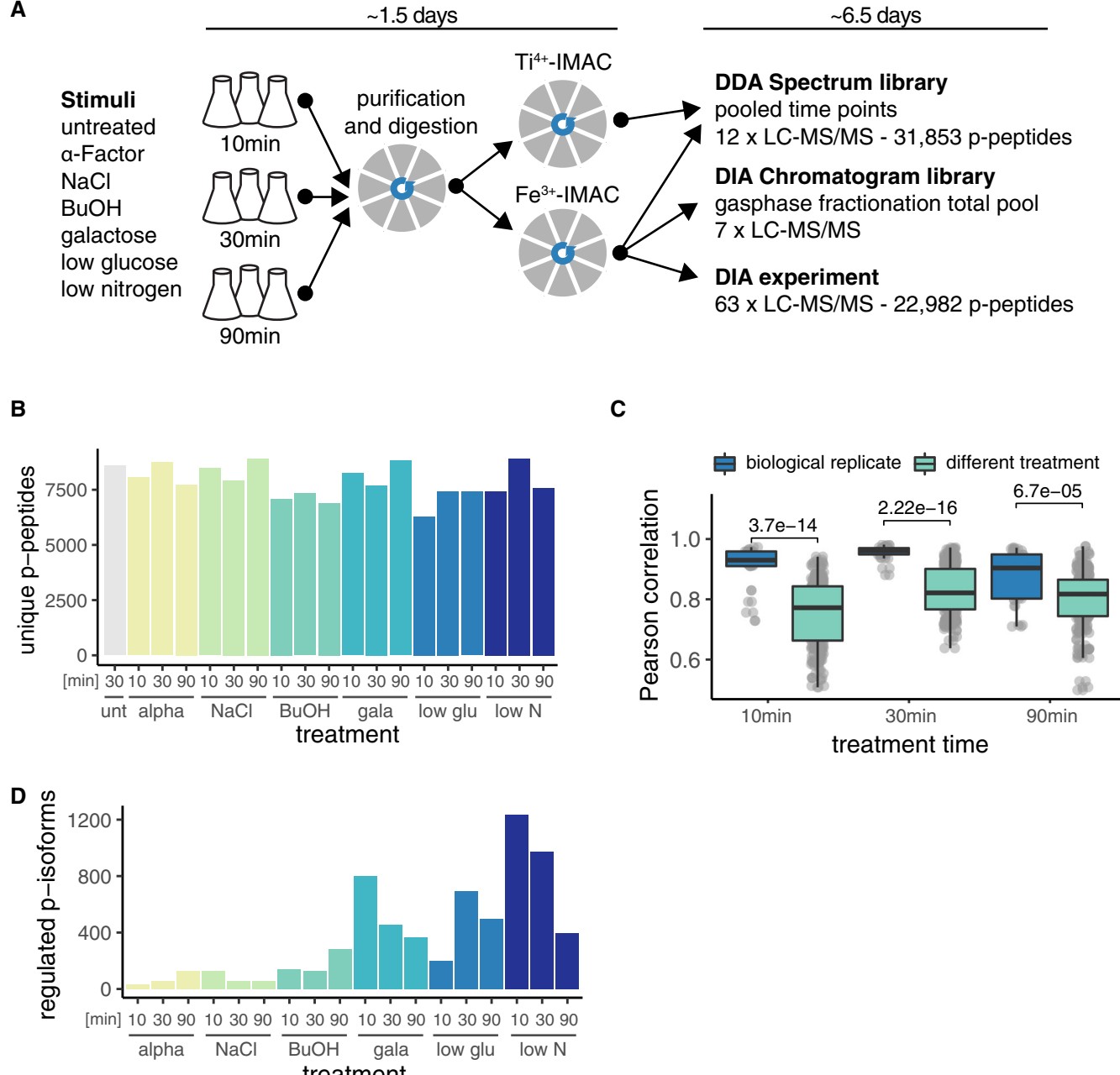

**Figure 4.  Multiple-perturbation phosphoproteomic experiment using R2-P2 and DIA-MS.**

A   Scheme of the experimental setup. Σ1278b MATa yeast cultures were grown and treated in biological triplicates with the indicated stimuli in a three-point time course. Proteins were purified and digested using R2-P2 with $Fe^{3+}$-IMAC or $Ti^{4+}$-IMAC. The spectrum library was generated by DDA measurement of pooled time points of $Fe^{3+}$-IMAC and $Ti^{4+}$-IMAC-enriched samples. The chromatogram library was generated by measuring a pooled sample from $Fe^{3+}$-IMAC enrichment of all conditions and time points in seven staggered narrow-window DIA-MS runs. The $Fe^{3+}$-IMAC fractions of each sample were measured in a single-injection wide-window DIA-MS experiment. The time used for sample preparation and measurement is indicated on the top and numbers of identified phosphopeptides on the right.

B   Unique phosphopeptides present in at least two (out of three) replicates identified in the DIA-MS experiment.

C   Boxplot depicting distributions of Pearson correlations for biological replicates and different treatments for different time points. *P*-values are calculated by Wilcoxon rank-sum test.

D   Significantly regulated phosphopeptide isoforms for the different stimulation and time points compared with the corresponding untreated control as determined by a two sample *t*-test (permutation-based FDR < 0.05, and fold change > 1.5). Only phosphopeptide isoforms with all phosphosites localized were considered.

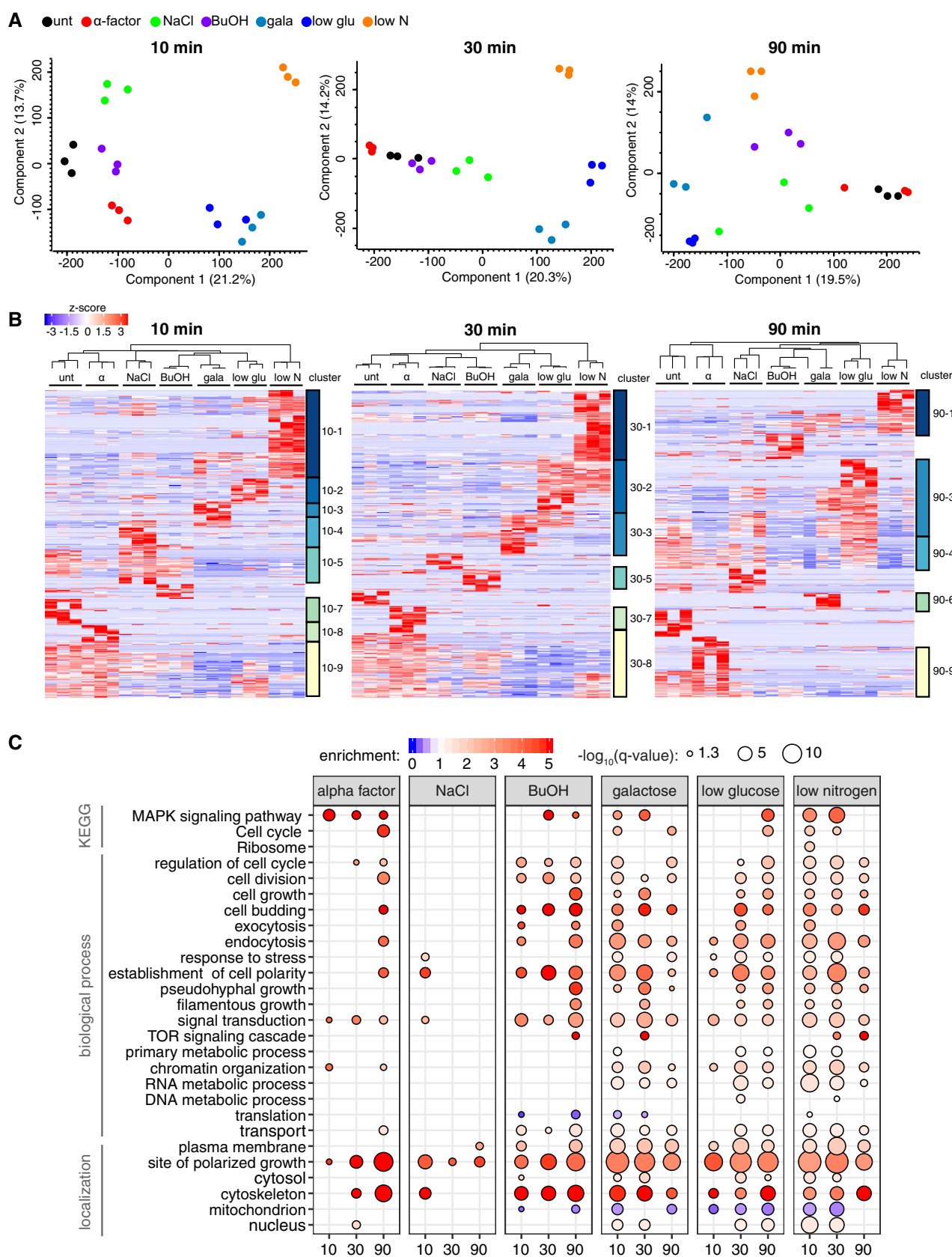

Figure 5.

**Figure 5.  Comparison of regulated phosphorylation across perturbations.**

A  Principal component analysis (PCA) of the DIA-MS phosphoproteomic data for the different stimuli separated by time points.
B  Hierarchical clustering analysis of z-scored intensity values of significantly regulated (ANOVA, permutation-based FDR < 0.05) phosphopeptides of the different time points. Column clustering hierarchy is indicated at the top. Row clusters are color coded and indicated on the right.
C  KEGG terms, biological process, and cellular component GO-terms associated with regulated phosphorylation sites (two sample *t*-test, untreated as control, permutation-based FDR < 0.05, and fold change > 1.5) for the different treatment and time points. Size of bubble denotes more significant enrichment/depletion of GO-terms (Fisher's exact test with Benjamini–Hochberg multiple-hypotheses corrected FDR < 0.02, the enrichment was calculated over the whole yeast proteome).

MAPK signaling pathway. Finally, all nutrient conditions showed enrichment of regulated phosphorylation event on proteins involved in chromatin organization and RNA metabolic pathways.

**Pathway analysis reveals stimulation-specific signatures on regulatory proteins and transcription factors**

To identify activated kinases and pathways, we inspected our data for changes in phosphorylation-based sentinels, i.e., phosphopeptide markers that are indicative of the activity of pathways or biological processes (Soste *et al*, 2014). We found evidence for activation of Pka and Snf1 kinases in nitrogen and glucose limitation, galactose, and butanol treatments (Fig EV7) indicative of activation of invasive growth. Given that developmental programs such as mating and invasive growth require alterations in cell cycle control, we expected cell cycle-specific phosphopeptide markers to be regulated under the tested conditions. Indeed, alpha factor downregulated activity markers of the cell cycle kinases Cdc28 and Ck2 (Fig EV7), indicative of G1 arrest to prepare for cell–cell fusion. We also found that Sch9 T723/S726, a sentinel of TORC1 activity, was suppressed under all nutrient limitations, and transiently reduced with NaCl (Figs EV7 and EV8), as previously reported (Urban *et al*, 2007).

To further investigate the effect of the treatments on TORC1 signaling, we mapped regulated phosphosites on a previously reported set of TORC1 components and effectors (Oliveira *et al*, 2015). We found multiple phosphosites on the TORC1 subunit Tco89 being regulated upon nitrogen and glucose starvation and to a lesser extent with galactose. Similarly, we found phosphosites on several regulatory proteins (Nap1, Aly2, Eap1, Igo1, Atg13) and transcription factors (Tod6, Dot6, Stb3, Fhl1, Ifh1, Maf1, Msn2, Msn4) downstream of TORC1 that were regulated under nutrient limitation (Fig EV8). The transcriptional repressors Maf1, Tod6, Dot6, and Stb3 regulate ribosomal biogenesis and are directly controlled by inhibitory phosphorylation through the nitrogen-sensitive TORC1-Sch9 and the carbon-sensitive PKA axis, thereby adjusting protein synthesis to nutrient availability (Huber *et al*, 2011). As expected, we see canonical Sch9 sites selectively downregulated (Maf1 S90, Tod6 S280, Stb3 S285, Stb3 S286) upon low nitrogen (Huber *et al*, 2009, 2011) and upregulated under low glucose, but other sites on the same proteins showed opposite trends. This differential regulation can be explained by varying activity of the TORC1-Sch9 axis, the opposing TORC1-PP2A axis, Pka, or another unknown pathway.

We mapped identified phosphorylation sites to the MAPK pathway from KEGG (Fig 6). The canonical HOG pathway consists of two MAPK branches converging upon the MAPKK Pbs2, which phosphorylates the MAPK Hog1. Activated Hog1 is imported to the nucleus and mediates the upregulation of ~600 genes via phosphorylation and/or interaction with various transcriptional activators and repressors including Msn2, Msn4, and Sko1 (Westfall, 2004). Here, we could not observe a distinct phosphorylation signature in the osmotic stress MAPK cascade for the high salt condition, possibly due to the transient nature of Hog1 activation peaking at 30–60 s (Kanshin *et al*, 2015) and/or the salt-induced response overlapping with other stimuli (Vaga *et al*, 2014). However, we found that phosphorylation of known effectors of the pathway was upregulated after salt stress (Msn2, Msn4, Sko1, Sic1, Hsl1, Hsl7; Fig 6).

In contrast, with alpha factor stimulation we found stimuli-specific, and often sustained, induction of phosphosites on the kinase Ste20, the pheromone-responsive MAPK scaffold Ste5, the MAPKKK Ste11 and the MAPK Fus3, the adaptor Ste50, and the downstream effectors Ste12, Dig1, Dig2, and Tec1 (Fig 6). Fus3 activates the mating response and suppresses the invasive growth (Breitkreutz & Tyers, 2002) by phosphorylating and tightly regulating the abundance of transcription factor Tec1 (Breitkreutz *et al*, 2001; Chou *et al*, 2004). As expected, we observe co-occurring phosphorylation at T273 and T276, a known phosphodegron (Bao *et al*, 2010), for all alpha factor stimulation time points. However, we also see this Tec1 dual phosphorylation at 10-min glucose limitation and singly phosphorylation of T273 at early time points of nitrogen starvation, indicating that nutrient limitation temporarily suppresses invasive growth. Tec1 activity and stability are regulated on its C-terminus by other not well-understood mechanisms (Köhler *et al*, 2002; Heise *et al*, 2010) including regulation through TORC1 (Brückner *et al*, 2011). We found that dynamic C-terminal phosphorylation at S325 opposes T273/T276 phosphorylation, and we postulate this site could be activating Tec1 by either stabilizing the protein, promoting its nuclear localization, or mediating binding to DNA or to other transcriptional machinery.

Taken together, our multiple-perturbation experiment identified informative and robust phosphorylation events on regulatory proteins and transcription factors, as well as effectors where multiple MAPK pathways converge such as Tec1, Msn4, and Stb3. Higher density time course experiments can bring additional resolution to the dynamic response of the MAPK signal transduction cascade, particularly for branches where phosphorylation is transient.

# Discussion

In this work, we have established R2-P2, a novel, automated phosphoproteomic sample preparation method that uses magnetic beads for both protein and peptide cleanup and phosphopeptide enrichment. R2-P2 is high-throughput (can handle 96 samples in parallel), rapid (1.5 days from cell lysate to proteomic and phosphoproteomic MS injections), robust (compatible with a variety of lysis buffers

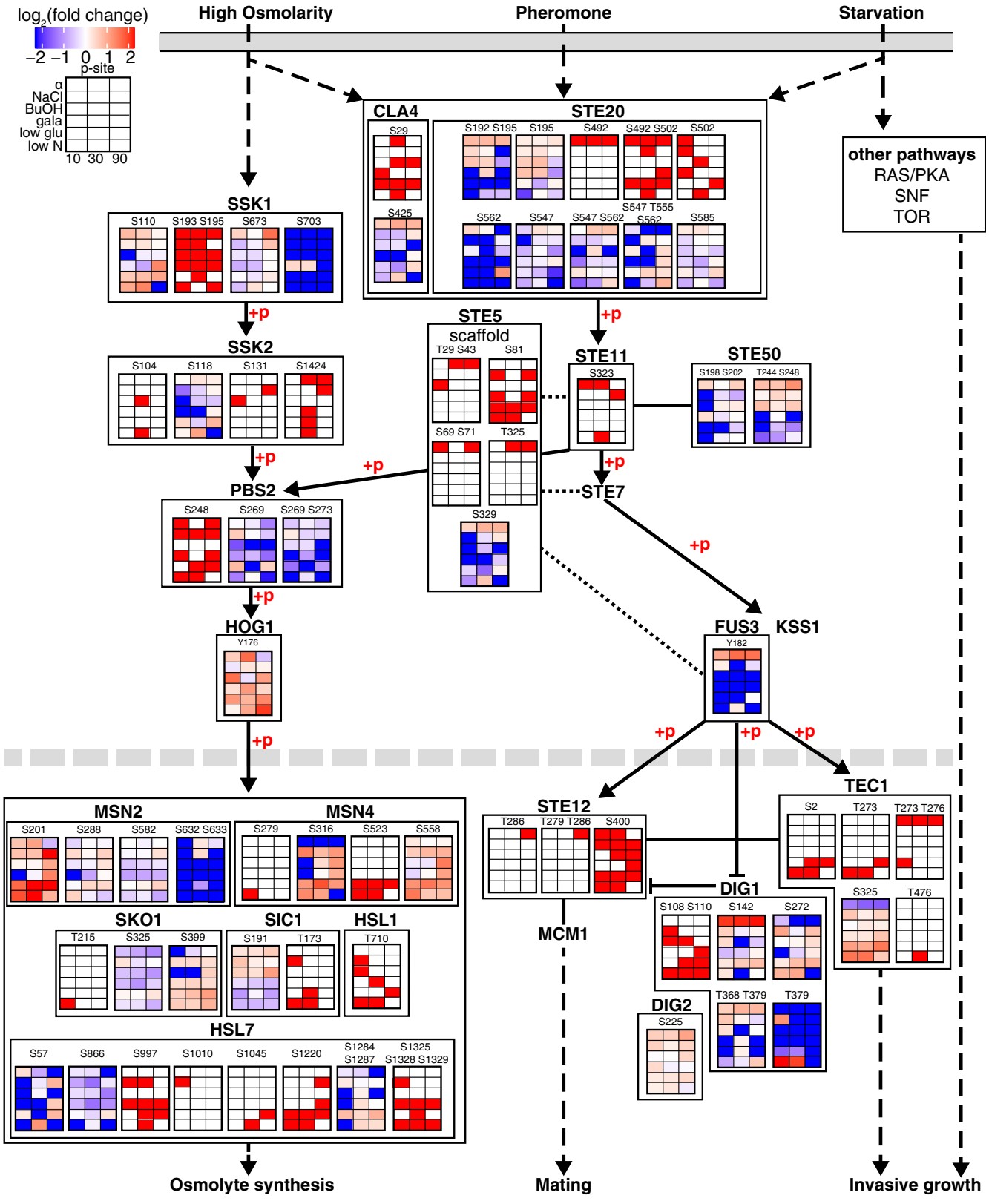

**Figure 6.  Phosphoregulation in proteins of the canonical MAPK pathways.**

Representation of partial mating, high-osmolarity, and invasive growth MAPK KEGG pathway from the plasma membrane to the nucleus. Each heatmap displays log₂ fold intensity changes over untreated of one or multiple co-occurring localized phosphorylation site(s) belonging to the indicated proteins. Heatmap rows correspond to treatments and columns to time points. Kinase substrate relationships (+p) and activating or inhibiting effect as stated by KEGG are indicated.

and detergents), reproducible (coefficient of variation for label-free phosphopeptide quantifications < 25%), high-fidelity (phosphopeptide enrichment selectivity > 90%), high sensitivity (~4,000 phosphopeptides from 25 μg yeast lysate), and cost-effective (< $3.50 for 250 μg protein). R2-P2 showed comparable results to in-solution digestion combined with SPE-C18 cleanup, judged by the number of identifications and quantitative reproducibility. As a note of caution, protease and phosphatase inhibitors are not efficiently removed by R2-P2 and can lead to significant inhibition of protein digestion or phosphopeptide enrichment, respectively. We have shown that R2-P2 is versatile with regard to the phosphopeptide enrichment chemistry. Fe-IMAC provides the best results in terms of total phosphopeptide intensity and identifications. Yet, we find that the combination of different materials can increase the coverage of the phosphoproteome. This could be attributed to the stochastic nature of data-dependent acquisition mass spectrometry, or the beads limited binding capacity and/or incomplete elution in batch enrichment regime, as suggested before (Ruprecht *et al*, 2015). Even though we show here the performance of R2-P1 and R2-P2 in label-free quantitative DDA and DIA measurements, the sample input levels we tested are also well-aligned with the requirements for the commonly used isobaric-tagging approaches. To facilitate implementation of the R2-P2 protocol in other laboratories, we provide a detailed protocol (Appendix) and downloadable programs (Dataset EV4 and Dataset EV5) for the magnetic bead processor.

Even though the yeast MAPK pathway is one of the best understood signaling systems in biology, our knowledge of signal integration and crosstalk with other pathways remains superficial. Global and pathway-centric analysis of phosphorylation changes induced by six different perturbations at three different time points reveals a network of interlocking events, rather than a set of simple linear pathways. This is evidenced by both phosphosites regulated under multiple perturbations (phosphosite-level crosstalk) and proteins with multiple phosphosites that are differentially regulated (protein-level crosstalk). Interestingly, MAPK branches feature different signaling dynamics with regard to their induction and recovery. Pathway-centric analysis recapitulated selective activation of the canonical HOG and mating MAPK pathways. Responses to nutrient starvation were more convoluted, likely due to signal inputs from additional kinases (e.g., Pka and Snf1). Despite the limited temporal exploration presented, we were able to observe robust phosphorylation of transcription factors and regulatory proteins; however, the temporal dynamics of the MAPK pathways shall be better recapitulated with additional early time points and rigorous modeling.

The R2-P1 and R2-P2 methods presented here will be attractive for any proteomic laboratory dealing with large sample batches, limited sample material, and challenging biological specimens. We have shown the advantages of R2-P2 for global phosphoproteomic studies; however, the method can be extended to automate the enrichment of peptides harboring other post-translational modifications, provided that enrichment chemistries can be added on to magnetic beads. For example, we anticipate future implementations of R2-P2 for peptide immunoaffinity enrichment of acetylation, ubiquitination, tyrosine phosphorylation, and other modifications or motifs. Due to its high sample capacity and robustness, R2-P2 can become a cornerstone method to study protein phosphorylation and regulatory signaling networks in biology and disease.

# Materials and Methods

**Reagents and Tools table**

| Reagent/Resource | Reference or Source | Identifier or Catalog Number |
|---|---|---|
| **Experimental Models** | | |
| BY4741 (*S. cerevisiae*) | Villen Lab | |
| Σ1278b MATa (*S. cerevisiae*) | Villen Lab | |
| **Chemicals, Enzymes and other reagents** | | |
| Sera-Mag SpeedBead Carboxylate-Modified Magnetic Particles (Hydrophilic) | GE Life Sciences | 45152105050250 |
| Sera-Mag SpeedBead Carboxylate-Modified Magnetic Particles (Hydrophobic) | GE Life Sciences | 65152105050250 |
| Fe-NTA MagBeads | Cube Biotech | 31501-Fe |
| $TiO_2$ microspheres | MagReSyn | MR-TID002 |
| Ti-IMAC microspheres | MagReSyn | MR-TIM002 |
| Zr-IMAC microspheres | MagReSyn | MR-ZRM002 |
| Water LC-MS grade | Fisher Scientific | 7732-18-5 |
| Acetonitrile LC-MS grade | Fisher Scientific | 75-05-8 |
| Methanol LC-MS grade | Fisher Scientific | A456-4 |
| Ethanol 200 Proof (100%) | Decon Labs | 2701 |
| Formic acid LC-MS grade | Fisher Scientific | A117-50 |

**Reagents and Tools table** (continued)

| Reagent/Resource | Reference or Source | Identifier or Catalog Number |
|---|---|---|
| Trifluoroacetic acid LC-MS grade | Fisher Scientific | A116-50 |
| Acetic acid glacial LC-MS grade | Fisher Scientific | A35-500 |
| Trichloroacetic Acid | Fisher Scientific | BP555-1 |
| Ammonia (ammonium hydroxide), $NH_4OH$ | Sigma-Aldrich | 221228-A |
| Ammonium bicarbonate | Sigma-Aldrich | A6141 |
| Ferric chloride, FeCl3 | Sigma-Aldrich | F2877 |
| EDTA | Sigma-Aldrich | EDS |
| Sequencing grade modified trypsin | Promega | V5111 |
| α-Factor Mating Pheromone | GenScript | RP01002 |
| 1-Butanol | Fisher Scientific | 02-002-038 |
| C8 Extraction Disks 3M™ Empore™ | Fisher Scientific | 14-386 |
| Pierce™ BCA Protein assay | Thermo Fisher | 23225 |
| **Software** | | |
| Comet | Eng *et al* (2013) | |
| Percolator | Käll *et al* (2007) | |
| Ascore | Beausoleil *et al* (2006) | |
| MSConvert | Chambers *et al* (2012) | |
| Skyline (version 4.2.0.19009) | MacLean *et al* (2010) | |
| EncyclopeDIA (version 0.8.1) | Searle *et al* (2018) | |
| Thesaurus (version 0.6.4) | Searle *et al* (2019) | |
| R | https://www.r-project.org/ | |
| Perseus | Tyanova *et al* (2016) | |
| **Other** | | |
| KingFisher™ Flex | Thermo Fisher Scientific | |
| KingFisher 96 KF microplate (200 μl) | Fisher Scientific | 22-387-030 |
| KingFisher 96 microtiter DW plate | Fisher Scientific | 22-387-032 |
| KingFisher 96 tip comb for DW magnets | Fisher Scientific | 22-387-029 |
| SpeedVac Vacuum Concentrator | Various | |
| Plate or PCR tube centrifuge | Various | |
| Q Exactive Hybrid Quadrupole-Orbitrap mass spectrometer | Thermo Fisher Scientific | |
| Q Exactive Plus Hybrid Quadrupole-Orbitrap mass spectrometer | Thermo Fisher Scientific | |
| Orbitrap Fusion Lumos Tribrid Mass Spectrometer | Thermo Fisher Scientific | |
| Easy1000 nanoLC | Thermo Fisher Scientific | |
| Easy1200 nanoLC | Thermo Fisher Scientific | |
| nanoACQUITY nanoLC | Waters | |

## Methods and Protocols

### Cell culture and treatment

For method optimization experiments, we used *S. cerevisiae* strain BY4741 and for stimulation experiment strain Σ1278b MATa. Yeast were cultured overnight at 30°C in synthetic complete (SC) medium with 2% glucose. Cells were subsequently inoculated at $OD_{600} = 0.1$ and grown to $OD_{600} = 0.6$. Cells were harvested by adding 100% trichloroacetic acid (w/v) to a final concentration of 10% trichloroacetic acid, incubated on ice for 10 min, centrifuged, decanted, washed once with 100% ice-cold acetone, centrifuged, acetone was decanted, pellets were snap-frozen, and stored at −80°C.

The following media were used to treat the Σ1278b strain: 20 mM alpha factor in SC-2% glucose, SC-0.2% glucose, 0.4 M NaCl in SC-2% glucose, 1% 1-butanol in SC-2% glucose, SC-2% galactose, and SLAD low nitrogen media (0.17% YNB, 50 μM ammonium sulfate, 2% glucose, leucine, uracil, tryptophan, histidine). To treat Σ1278b, cells were grown to OD600 = 0.6, harvested by centrifugation, and washed twice with pre-warmed fresh media base (SC-2% glucose, SC-0.2% glucose, SC-2% galactose, or SLAD), and then,

treatments were induced for 10, 30, and 90 min followed by trichloroacetic acid harvest as described above.

### Cell lysis, protein reduction, and alkylation

Frozen cell pellets were resuspended in lysis buffer composed of 8 M urea, 150 mM NaCl, and 100 mM Tris pH 8.2. Cells were lysed by 3 cycles of bead beating (30-s beating, 1-min rest) with zirconia/silica beads. Lysate protein concentration was measured by BCA assay. Proteins were reduced with 5 mM dithiothreitol (DTT) for 30 min at 55°C and alkylated with 15 mM iodoacetamide in the dark for 15 min at room temperature. The alkylation reaction was quenched by incubating with additional 5 mM DTT for 15 min at room temperature.

### R2-P1 protocol

The R2-P1 purification and digestion were implemented in the following way on the KingFisher™ Flex: The 96-well comb is stored in plate #1, magnetic carboxylated beads in plate #2, lysate-ethanol mixture in plate #3, wash solutions in plates #4 to #6, elution/digestion buffer with the digestion enzyme in plate #7, and second elution (water) in plate #8 (Fig 1A).

The method was configured to collect the carboxylated beads in plate #2, move them to plate #3 for protein binding and subsequently to plate #4, #5, and #6 for protein cleanup. The protein cleanup takes 30 min, and the protocol pauses before loading of plate #7 containing the digestion enzyme, which allows for preparation of the digestion solution only immediately before use. The beads are then moved to plate #7, and proteins are eluted/digested at 37°C for 3.5 h with constant agitation. Carboxylated beads are subsequently moved and washed in the second elution plate #8 and afterward discarded. At this step, the robotic program ends and plates can be removed. The two elution plates (#7 and #8) are combined in one plate, acidified with formic acid, and clarified by centrifugation. At this point, aliquots are taken for total proteome analysis. The rest of the peptides are dried down and can be taken for R2-P2. For the comparison experiments in this study, the different conditions were processed together in the same R2-P1 run, unless otherwise stated.

### Important notes

- Protease inhibitors: We have seen that Pierce protease inhibitors (#A32963) are not efficiently removed by R2-P1 and lead to decreased digestion efficiency by trypsin or LysC if they are present at > 0.2× in the digestion buffer. We recommend using denaturing buffers (e.g., 8 M urea) without protease inhibitors for lysis.
- Phosphatase inhibitors: We have seen that a common phosphatase inhibitor mix (50 mM sodium fluoride, 10 mM sodium pyrophosphate, 50 mM sodium beta-glycerophosphate, 1 mM sodium orthovanadate) is not efficiently removed by R2-P1, leading to competition during phosphopeptide enrichments. We could not observe any negative effect of phosphatase inhibitors on LC-MS/MS analysis of total proteome. If phosphopeptide enrichment is planned, we recommend using denaturing buffers (e.g., 8 M urea) without phosphatase inhibitors for lysis.
- Precipitates: Depending on the sample, we have observed insoluble particles due to precipitation at a few steps in the protocol. It is critical that phosphopeptide enrichment is performed on a clarified peptide mixture for maximal efficiency. If precipitation is

observed at any step after protein digestion, the plate needs to be centrifuged for 10 min at maximal speed and supernatant transferred to a new plate. If this does not help to remove visible precipitates, longer centrifugation is needed. Alternatively, samples can be transferred to PCR tube strips and centrifuged in a PCR tube centrifuge at higher speeds.

- All steps are performed at room temperature, unless stated otherwise.
- The following protocol is for 250 µg of protein per well. The protocol has been successfully tested for protein input amounts from 25 to 500 µg.
- The protocol has been most extensively tested with lysates in urea buffer (8 M urea, 150 mM NaCl, 100 mM Tris pH 8); however, other lysis buffers should work as well.
- With the recommended buffer volumes and ratios, the maximal amount of protein that can be processed in one well is 500 µg.
- We perform digests for 3.5 h at 37°C; however, these parameters can easily be adjusted as needed in the protocol.
- All plate pipetting steps are performed with 8 or 12 multi-channel pipettes.

### Experiment planning

1 Determine the amount of protein sample that needs to be processed, considering:
  a. For proteome analysis only, we recommend starting with 25 µg of protein.
  b. For phosphoproteomic analysis, we recommend starting with a minimum of 100 µg protein, with an optimal starting amount of 200–500 µg. This is sample dependent and needs to be tested.
2 Determine the protein concentration of clarified cell lysate(s) by BCA assay or equivalent. Reduce and alkylate cysteines by preferred method. Adjust protein concentration in samples to 1 µg/µl with lysis buffer.
3 Determine the amount of magnetic carboxylated beads to be used. We recommend using 1 µl of 10 µg/µl of carboxylated bead mix per µg of protein to be processed.
4 Determine the volumes of the solutions to be used per well in the different plates:
  a. Binding plate:
    i. Lysate volume ($V_{Lysate}$) = Calculate lysate volume to reach desired protein amount, keeping a protein concentration of 1 µg/µl.
    ii. 100% EtOH volume = $V_{Lysate}$ (to reach 50% EtOH v/v).
  b. Wash plates (3×): 80% EtOH volume = $2 \times V_{Lysate}$
  c. Elution-1 plate:
    i. 150 µl 25 mM ammonium bicarbonate pH 8.2
    ii. Trypsin or LysC: 1 µg enzyme: 100 µg protein
  d. Elution-2 plate: 100 µl water
5 Determine the type of plates to be used for the different solutions: For volumes of 50–150 µl per well, use Kingfisher microplates (shallow well), and for 150–1,000 µl per well, use Kingfisher microtiter DW plates (deep well).

### Carboxylated magnetic bead preparation

1 Take both (hydrophilic and hydrophobic) magnetic carboxylated bead stocks off the fridge, warm to room temperature, and vortex gently to fully suspend magnetic beads.

2  Beads come at a stock concentration of 50 μg/μl. Take the required amount of beads and mix hydrophilic and hydrophobic beads at a 1:1 ratio. Dilute to a total bead concentration of 1 μg/μl, wash the beads three times with water keeping the beads at 1 μg/μl on a magnetic rack in an Eppendorf or Falcon tube, and resuspend beads in water at the working concentration of 10 μg/μl.

### R2-P1 procedure

1  Prepare an empty plate containing the tip comb.

2  Prepare the bead plate by adding the calculated amount of beads to the plate. If the volume is < 50 μl, fill up to 50 μl with water.

3  Transfer the calculated volumes of cell lysate (1 μg/μl) to the binding plate.

4  Add 100% EtOH to the binding plate to reach 50% EtOH (v/v) (no mixing or pipetting required).

5  Prepare the three wash plates by dispensing 80% EtOH.

6  Prepare Elution-2 plate by dispensing water.

7  Start the R2-P1 Kingfisher program and follow the instructions on the robot. Leave the space for Elution-1 plate empty and start the protocol.

8  Start preparing Elution-1 plate on ice 25 min into the protocol.

9  The R2-P1 Kingfisher program will pause after approximately 35 min. Follow the robot's instructions, load Elution-1 plate, and resume the program.

10  Remove all plates after the program finishes.

11  Transfer solution in Elution-2 plate to Elution-1 plate.

12  Stop the enzymatic digestion by adding formic acid to the digest to pH < 2 (usually final formic acid concentration is 3–5%).

13  Check for precipitates, if observed any, remove them by centrifugation and transfer the supernatant to a new plate.

14  Take a sample aliquot for total proteome analysis, see optional step 15 or transfer to a MS vial or MS sample plate.

15  (Optional step that we recommended when processing many samples in parallel): To avoid any carryover of magnetic beads to the LC system, filter samples through C8 stage tips or stage tip plate. For this, pack stage tips with one layer of C8 material and perform the following steps:
   a.  Condition with 30 μl MeOH.
   b.  Wash with 30 μl 100% ACN.
   c.  Wash with 30 μl 70% ACN, 0.25% acetic acid.
   d.  Hold stage tip on top of MS sample vial.
   e.  Adjust sample to 50% ACN and pass sample through stage tip, collecting eluate in the MS vial.
   f.  Elute with 30 μl 70% ACN, 0.25% acetic acid, collecting this second eluate also in the MS vial.

16  Dry samples down in a SpeedVac. Dried samples can be stored at −20°C. Resuspend in 4% formic acid and 3% ACN prior to LC-MS analysis. The rest of the plate is dried down and can be stored at −20°C for subsequent phosphopeptide enrichment.

### R2-P2 protocol

The automated phosphopeptide enrichment was implemented in the following way on the KingFisher™ Flex (Thermo Scientific): The 96-well comb is stored in plate #1, magnetic $Fe^{3+}$-IMAC, $Ti^{4+}$-IMAC, $Zr^{4+}$-IMAC, or $TiO_2$ beads in plate #2, resuspended peptides in plate #3, wash solutions in plates #4 to #6, and elution solution in plate #7 (Fig 1A). Shallow 96-well KingFisher plates were used.

The method was configured to collect the magnetic beads in plate #2, move them to plate #3 for phosphopeptide binding and subsequently to plate #4, #5, and #6 for washing. The phosphopeptide purification takes 40 min, and the protocol pauses to allow for loading of plate #7 containing the elution solution, which is prepared immediately before use, to avoid evaporation of ammonia. The beads are subsequently moved to plate #7 where phosphopeptides are eluted. Plates are removed from the robot at this point, and the elution is immediately neutralized by acidification. Peptides are dried down and stored at −20°C until MS analysis. For the comparison experiments in this study, the different conditions were processed together in the same R2-P2 run, unless otherwise stated.

### Important notes

- Magnetic particles for phosphopeptide enrichment: The phosphopeptide enrichment described in this protocol is performed using $Fe^{3+}$-IMAC magnetic particles. However, we have also successfully tested R2-P2 with $Ti^{4+}$-IMAC, $TiO_2$, and $Zr^{4+}$-IMAC microspheres from MagReSyn following the manufacturer's instructions. The results were of comparable quality and different resins provided orthogonal phosphopeptide identification.
- Any precipitate present in the peptide sample significantly reduces the selectivity of phosphopeptide enrichment and needs to be removed by centrifugation prior to the binding step.
- Do not leave magnetic beads without liquid for longer than 1 min.
- Do not leave magnetic beads in aqueous solutions for longer than 2 h.
- If magnetic beads are clumped or aggregated, sonicate them briefly in a bath sonicator.
- Magnetic $Fe^{3+}$-IMAC beads can be reused by stripping the ion metal and reloading it. We have observed no change in performance after reuse for up to 1 year.
- All plate pipetting steps are performed with 8 or 12 multi-channel pipettes.

### Experiment planning

1  Determine the amount of magnetic $Fe^{3+}$-IMAC beads to be used. We recommend using 2.5 μl of 5% beads per 10 μg of peptides. However, this is sample dependent and might need to be adjusted.

2  Determine the volumes of the solutions to be used per well in the different plates:
   a.  Binding plate: 150 μl 80% ACN, 0.1% TFA
   b.  Wash plates (3×): 150 μl 80% ACN, 0.1% TFA
   c.  Elution plate: 50 μl 50% ACN, 2.5% $NH_4OH$
   d.  Neutralization solution for elution plate: 30 μl 75% ACN, 10% formic acid

### Usage of new $Fe^{3+}$-NTA MagBeads

1  PureCube Fe-NTA MagBeads are delivered as a 25% suspension and are ready to use for phosphopeptide enrichment.

2  Before use, dilute beads to 5% and wash three times with 80% ACN, 0.1% TFA. For all further handling steps, beads are kept at 1 ml aliquots at a working concentration of 5%.

### Storage of used Fe³⁺-NTA MagBeads

1  After R2-P2, beads can be recollected before they dry out.
2  Wash once with 1 ml 50% ACN, 50% MeOH, 0.01% acetic acid.
3  Store beads in 1 ml of the same buffer at 4°C until further use.

### Stripping and reloading of Fe³⁺-NTA MagBeads

1  Wash beads three times with 1 ml of water.
2  Wash once with 1 ml 40–100 mM EDTA, pH 8.
3  Resuspend beads in 1 ml 40–100 mM EDTA, pH 8 and incubate for 30 min while shaking or rotating the tubes. Ensure that the beads remain in solution.
4  Wash beads three times with 1 ml of water.
5  Wash once with 1 ml 10 mM $FeCl_3$.
6  Resuspend in 1 ml 10 mM $FeCl_3$ and incubate for 30 min, while shaking or rotating in tubes. Ensure that the beads remain in solution.
7  Wash beads three times with 1 ml of water.
8  Wash beads three times with 1 ml 80% ACN, 0.1% TFA.
9  Resuspend in 1 ml 80% ACN, 0.1% TFA.
10  Beads are ready to use.

### Fe³⁺-IMAC procedure for R2-P2

1  Resuspend dried peptides from R2-P1 directly in the plate with 150 µl 80% ACN, 0.1% TFA by shaking and incubating in a bath sonicator for 10 min.
2  Make sure that peptides go completely into solution and there is no precipitate. Insoluble precipitates need to be removed by centrifugation.
3  Prepare an empty plate containing the tip comb.
4  Prepare one plate with the calculated amount of IMAC beads. If the volume is < 50 µl, fill up to 50 µl with 80% ACN, 0.1% TFA.
5  Prepare three wash plates with 150 µl 80% ACN, 0.1% TFA per well.
6  Start the R2-P2 Kingfisher program and follow the instructions on the robot. Leave elution plate position empty and start the protocol. The elution plate is prepared and loaded later in order to prevent evaporation of the $NH_4OH$, which could compromise elution efficiency.
7  The R2-P2 Kingfisher program will pause after approximately 35 min. At this point, pipette 50 µl of 50% ACN, 2.5% $NH_4OH$ into each well of the elution plate.
8  Follow robot instructions, load the elution plate and resume the program.
9  Remove all plates after the program finishes.
10  Immediately neutralize the elution by adding 30 µl 75% ACN, 10% formic acid to the plate.
11  See optional step 12 or transfer to a MS vial or MS sample plate.
12  (Optional step that we recommended when processing many samples in parallel): To avoid any carryover of magnetic beads to the LC system, filter samples through a C8 stage tip or stage tip plate. For this, pack a stage tip with one layer of C8 material and perform the following steps:
    a.  Condition with 30 µl MeOH.
    b.  Wash with 30 µl ACN.
    c.  Wash with 30 µl 70% ACN, 0.25% acetic acid.
    d.  Hold stage tip on top of MS sample vial.
    e.  Adjust sample to 50% ACN and pass sample through stage tip, collecting eluate in the MS vial.
    f.  Elute with 30 µl 70% ACN, 0.25% acetic acid, collecting this second eluate also in the MS vial.
13  Dry samples down in a SpeedVac. Dried samples can be stored at −20°C. Resuspend in 4% formic acid and 3% ACN for LC-MS analysis.

### Mass spectrometry data acquisition

Dried peptide and phosphopeptide samples were dissolved in 4% formic acid, 3% acetonitrile and analyzed by nLC-MS/MS. Peptides were loaded onto a 100 µm ID × 3 cm precolumn packed with Reprosil C18 3 µm beads (Dr. Maisch GmbH), and separated by reverse-phase chromatography on a 100 µm ID × 30 cm analytical column packed with Reprosil C18 1.9 µm beads (Dr. Maisch GmbH) and housed into a column heater set at 50°C.

Method optimization experiments were performed on a Q Exactive or Q Exactive Plus Hybrid Quadrupole-Orbitrap mass spectrometer (Thermo Fisher) equipped with an Easy1000 or an Easy1200 nanoLC system, respectively. Phosphopeptides were separated by a 60-min gradient ranging from 7 to 28% acetonitrile in 0.125% formic acid. For DDA experiments, full MS scans were acquired from 350 to 1500 m/z at 70,000 resolution with fill target of 3e6 ions and maximum injection time of 100 ms. The 20 most abundant ions on the full MS scan were selected for fragmentation using 2 m/z precursor isolation window and beam-type collisional-activation dissociation (HCD) with 26% normalized collision energy. MS/MS spectra were collected at 17,500 resolution with fill target of 1e5 ions and maximum injection time of 50 ms. Fragmented precursors were dynamically excluded from selection for 30 s.

Data-dependent acquisition for spectral library generation, DIA for gas-phase fractionation, and DIA for quantification were performed on an Orbitrap Fusion Lumos Tribrid Mass Spectrometer (Thermo Fisher) equipped with a nanoACQUITY nanoLC (Waters). Phosphopeptides were separated by a 60-min gradient ranging from 7 to 28% acetonitrile in 0.125% formic acid. All MS spectra were acquired on the orbitrap mass analyzer and stored in centroid mode. For the DDA experiments, full MS scans were acquired from 350 to 1,500 m/z at 60,000 resolution with an AGC target of 7e5 ions and maximum injection time of 50 ms. The most abundant ions on the full MS scan were selected for fragmentation using 1.6 m/z precursor isolation window and beam-type collisional-activation dissociation (HCD) with 28% normalized collision energy for a cycle time of 3 s. MS/MS spectra were collected at 15,000 resolution with fill target of 5e4 ions and maximum injection time of 50 ms. Fragmented precursors were dynamically excluded from selection for 40 s. For the DIA gas-phase fractionation, the Fusion Lumos was configured to acquire seven runs, each spanning ~100 m/z (448.475–552.524, 548.525–650.574, 648.575–750.624, 748.625–850.674, 848.675–950.724, 948.725–1050.774, 1048.775–1150.824). Full MS scans were acquired in centroid mode from 440 to 1,160 m/z at 60,000 resolution, AGC target of 1e6, and maximum injection time of 100 ms. MS/MS spectra were acquired at 30,000 resolution, using 4 m/z precursor isolation window, AGC target of 2e5, maximum injection time of 54 ms, and HCD with 28% normalized collision energy. For quantitative DIA analysis, the Fusion Lumos was configured to acquire a full MS scan in centroid mode from 350 to 1,500 m/z at 60,000 resolution, AGC target 7e5 and maximum

injection time 50 ms followed by 48 × 15 m/z DIA spectra using 1 m/z overlapping windows from 472.489 to 1,145.489 m/z at a 15,000 resolution, AGC target 5e4, and maximum injection time of 22 ms.

### Mass spectrometry data analysis

DDA-MS/MS spectra were searched with Comet (2015.02.rev.5; Eng *et al*, 2013) against the *S. cerevisiae* proteome. The precursor mass tolerance was set to 20 ppm. Constant modification of cysteine carbamidomethylation (57.021463 Da) and variable modification of methionine oxidation (15.994914 Da) were used for all searches, and additional variable modification of serine, threonine, and tyrosine phosphorylation (79.966331 Da) was used for phosphopeptide samples. Search results were filtered to a 1% FDR at PSM level using Percolator (Käll *et al*, 2007). Phosphorylation sites were localized using an in-house implementation of the Ascore algorithm (Beausoleil *et al*, 2006). Phosphorylation sites with an Ascore > 13 ($P < 0.05$) were considered confidently localized. Peptides were quantified using in-house software measuring chromatographic peak maximum intensities.

For overlapping DIA runs, MSConvert (Chambers *et al*, 2012) was used to deconvolute RAW files and generate mzMLs. A spectrum library was created from the Fusion Lumos DDA data using Skyline (version 4.2.0.19009; MacLean *et al*, 2010). This BLIB library was imported to EncyclopeDIA (version 0.8.1) (Searle *et al*, 2018), used to search the DIA gas-phase fractionated runs and a chromatogram library was generated. The resulting ELIB file was imported to Thesaurus (version 0.6.4; Searle *et al*, 2019) to analyze the DIA files, obtaining peptide identifications, site localizations, and quantifications. In Thesaurus, we used the following settings: phosphorylation (STY) as modification type, non-overlapping DIA as data-acquisition type, recalibrated (peak width only) as localization strategy, B/Y ions for fragmentation, 60,000 resolution as precursor mass tolerance, 15,000 resolution as fragment mass tolerance, minimum of 3 well-shaped fragment ions, and Percolator (v3-01) threshold of < 1% FDR.

### Bioinformatic analysis

Bioinformatic analysis was performed using R (https://www.r-project.org/) and Perseus (Tyanova *et al*, 2016). Quantitative values obtained from DDA and DIA analysis were median normalized, unless otherwise stated.

For all boxplots, the lower and the upper hinges of the boxes correspond to the 25% and 75% percentile, and the bar in the box to the median. The upper and lower whiskers extend from the largest and lowest values, respectively, but no further than 1.5 times the IQR from the hinge.

For the DIA experiments, phosphopeptide isoforms (peptides containing the same combination of phosphosites) were required to be observed in 2 out of 3 biological replicates (same treatment, same time point); otherwise, the missing value was replaced with 0. For phosphosite-centric analyses (Figs 4D, 6, EV7, and EV8), only phosphopeptide isoforms were considered where all phosphosites could be localized based on site-specific fragment ions at least once and no conflicting positional isomers were localized in other samples.

All correlation calculations utilize Pearson's method. The calculation of individual *q*-values was performed using two-sided Student's *t*-tests of biological replicates, using the "untreated" condition as control and applying a permutation-based FDR < 0.05 and fold change > 1.5 filters, unless otherwise noted. To obtain data for hierarchical clustering and heatmap plots, ANOVA multiple-sample tests were performed on intensity values of phosphopeptides and data were filtered for values with Benjamini–Hochberg multiple-hypothesis-corrected *q*-value < 0.05. For hierarchical clustering, *Z*-score values were used as input, and for all other heatmaps, $\log_2$ fold changes over the untreated were used.

Principal component analysis (PCA) and GO term enrichment analyses were performed using Perseus. GO enrichment analysis was performed by annotating phosphopeptides with the corresponding protein annotation terms and using the whole yeast proteome as background. Fisher exact test with Benjamini–Hochberg multiple-hypothesis correction was applied at the protein level and filtered for FDR < 0.02. The categories used for the annotation of proteins were Gene Ontology (GO) Biological Processes, GO Cellular Components, and Kyoto Encyclopedia of Genes and Genomes (KEGG).

## Data availability

All mass spectrometry proteomics data have been deposited to the ProteomeXchange Consortium (http://www.proteomexchange.org/) via the PRIDE partner repository (Perez-Riverol *et al*, 2019) with the dataset identifier PXD013453.

**Expanded View** for this article is available online.

### Acknowledgements

We would like to thank Ian Smith for advice with the MS measurements, Anthony Valente for bioinformatic support, and all members of the Villén lab for useful discussions. We also thank Stoyan Stoychev (ReSyn Biosciences) for providing Ti$^{4+}$-IMAC, Zr$^{4+}$-IMAC, and TiO$_2$ beads and advice on how to use them. M.L. is supported by a postdoctoral fellowship from the Swiss National Science Foundation (P2ZHP3_181503). This work is supported by NIH grants R35GM119536 and R01AG056359, Human Frontiers Science Program grant RGP0034/2018, a Research program grant by the W.M. Keck Foundation, and the University of Washington's Proteomics Resource UWPR95794.

### Author contributions

ML, RAR-M, and JV conceived the study. ML conducted most of the experiments, with assistance from NKF and advice from RAR-M. ML analyzed the data. JV supervised the study. ML and JV wrote the paper and all the authors edited it.

### Conflict of interest

The authors declare that they have no conflict of interest.

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
