## [Review Process File · Molecular Systems Biology]

R2-P2 rapid-robotic phosphoproteomics enables multidimensional cell signaling studies

Mario Leutert, Ricard A. Rodriguez-Mias, Noelle K. Fukuda and Judit Villén.

Review timeline:

Submission date:	22 nd May 2019
Editorial Decision:	13 th June 2019
Revision received:	8 th October 2019
Editorial Decision:	29 th October 2019
Revision received:	14 th November 2019
Accepted:	18 th November 2019

Editor: Maria Polychronidou

Transaction Report:

1st Editorial Decision.

13th June 2019

Thank you again for submitting your work to Molecular Systems Biology. We have now heard back from the three referees who agreed to evaluate your study. As you will see below, the reviewers acknowledge that the presented approach is a potentially valuable contribution for the proteomics field. They raise however a series of concerns, which we would ask you to address in a major revision.

I think that the reviewers' recommendations are rather clear and there is therefore no need to repeat the comments listed below. Some of the more fundamental concerns are raised by reviewer #1 who mentions that further analyses need to be performed in order to better support the reproducibility of the results and the performance of the method. As reviewers #1 and #3 mention, some level of validation of the results obtained by the application of R2-P2 on yeast signaling would significantly enhance the impact of the study. While we agree with reviewer #1 that the inclusion of follow-up experiments is not mandatory since the main focus of the study is the presentation of the method, we would not be opposed to the inclusion of such analyses if you have already performed them or feel inclined to do so.

All other issues raised by the reviewers need to be satisfactorily addressed. As you may already know, our editorial policy allows in principle a single round of major revision so it is essential to provide responses to the reviewers' comments that are as complete as possible. Please feel free to contact me in case you would like to discuss in further detail any of the issues raised by the reviewers.

REFeree REPORTS

Reviewer #1:

In this manuscript, Leutert et al have implemented a robotic liquid handling procedure for sample preparation in a phosphoproteomics workflow. Specifically, they combine SP3, a previously developed method for protein clean up from cell extracts, with phospho-peptide enrichment in an integrated procedure on a KingFisher based on the use of magnetic beads in both parts of the workflow. The authors benchmark performance of the system compared to conventional methods, showing high reproducibility and scalability. They next apply the methodology to a phosphoproteomic analysis of yeast to probe response to various perturbations, using a data-independent mass spectrometric approach, and they perform pathway analysis to identify changes in phosphorylation events that are unique or shared between the treatments, denoting treatment-specific response or crosstalk. Authors conclude that the presented workflow should be broadly applicable for high-throughput applications under standardized conditions.

This is a well-designed study, addressing a general need in proteomics to enhance standardization in proteomic sample preparation to improve reproducibility and throughput. The presented procedure implemented on a KingFisher liquid handling robot is a highly relevant contribution in this direction, taking advantage of magnetic bead-based procedures for protein clean-up and phospho-peptide enrichment by SP3 and metal affinity beads, respectively. The latter type have been extensively used in the literature in various configurations the literature, however not on this type of robotic system. SP3 is a relatively novel method for generic protein clean-up now implemented on a robot for the first time. To assess performance of a platform as introduced here, both bead types should be evaluated separately and in combination with appropriate metrics to indicate reproducibility, variance, sensitivity etc. As detailed below, one chief concern is that the authors could have been somewhat more rigorous in scrutinizing performance. The application of the established workflow to various stresses in yeast is nice and relevant, showing that the R2-P2 can be effectively used to probe multiple samples to disclose new biology. Yet, none of this (e.g. pathway crosstalk) was further validated, however this may not be a strict requirement for a paper with a methods focus.

Specific comments:

1. Page 6: Assessing quantitative reproducibility is key to demonstrate robust performance of the presented R2P2 platform. However this was only shown for an experiment with just 3 replicates (Fig 1E), and it is unclear if these samples were prepared within one batch, or independently across batches, ideally over an extended period of time. Such data should be shown to more convincingly demonstrate stable performance with low variability.
2. For the SP3-based method, the number of identified phosphopeptides does not scale with the amount of input material (Fig 2A). In fact, only slightly more phosphopeptides were identified from 400ug starting material than from 25ug. Authors should clarify whether this is due to SP3 or phospho-enrichment, or rather sub-optimal interfacing of the two. Judging from expected scaling when using SepPack (left part of Fig 2A), the effect seems to be due to SP3. This begs for an explanation, or for more replicates (just 2 as shown may not suffice) to verify reproducible recovery of proteins from high-input samples by SP3, with and without subsequent phospho-enrichment. Furthermore it must be ruled out that variability is due to sub-optimal sample handling on the KingFisher, which normally speaking should outcompete manual processing by SepPack in terms of reproducibility. Also, it should be excluded that variance is due to the absence of phosphatase inhibitors in the SP3-based method, while these are present (?) when using SepPack.
3. The authors claim that R2-P2 has higher sensitivity than conventional systems (page 8 and abstract). It is not clear which observations lead to this conclusions, since scaling experiments were tested towards higher input, not lower input.
4. Page 9: Reproducible preparation of 96 samples is claimed, however data for only 3 replicates was presented. Therefore claims should be toned down, or more samples should be analysed.
5. In Fig S5A it is striking and slightly worrying that replicate samples are not always as closely spaced as one would wish. The authors claim that this is due to biological variation, especially for the 90-min conditions, however this was not substantiated. Since this is primarily a methodological paper aiming to establish a KingFisher-based workflow, it would be extremely useful to

experimentally demonstrate that technical and biological variability can be distinguished, and that the former is smaller than the latter. Therefore, authors should perform replicates from the same yeast isolate (e.g. one of the 90-min conditions) and demonstrate that process-replicates are more closely spaced than biological replicates in a PCA analysis.

6. It is unclear what can be gathered from the GO analysis in Fig S5B, since it includes many and diverse biological processes across the various perturbation conditions. Also it is unclear how the authors get to the conclusion that component 2 resolves temporal effects. It may be better to remove this figure.

7. Similarly for Fig5C: Given the diversity of GO-terms recovered between treatments or, unexpectedly, between time points, anything can be concluded from Fig 5C, and therefore at the same time nothing can be distilled. Really there is no individual or unifying conclusion that can be drawn from this analysis. To a large part this is also due to the fact that many terms are very general and therefore minimally informative (transport, response to stimulus, RNA metabolic process etc). Unsurprisingly, the text associated with this figure is highly descriptive recapitulating what one can read in the figure, without clarifying discrepancies or making biological inferences. It would have been more informative to select one treatment (with time points) and perform a more in-depth data analysis, or to leave out this figure altogether.

8. Fig5B: The basis is unclear for the general conclusion that the 90-min clusters include phosphosites regulated in multiple conditions, more so than the 10 and 30-min clusters: all 3 heatmaps show regions indicating generally changing peptides (lower sections for 10 and 30 min, middle section for 90 min).

Reviewer #2:

In this report by Leutert and colleagues, the authors develop and report on a semi-automated proteomic and phosphoproteomic sample processing workflow that leverages a combination of solvent-based protein aggregation on microparticles and magnetic robot-based automation to generate reproducible digests and purifications for downstream LC-MS/MS-based analysis. As the field of proteomics moves towards higher throughput and into less specialized laboratories, the need for smarter sample processing and handling increases, which frames the rationale and significance of the work as described.

This is a really well-written manuscript that describes (in almost too fine!) details of the conceptual basis for the microaggregation-based sample handling, the careful validation experiments, and the comparisons with existing technology that appear to make their "R2P1/P2" methods as good as the status quo. Given the increase in adaptation of isobaric tagging methods in quantitative proteomics and phosphoproteomics, the sample input levels as described in the paper are well aligned with the typical downstream applications for such methods (in spite of the DIA application in this manuscript). The figures accurately and clearly convey the message and the study seems well grounded. My personal opinion is that the authors could have pared down their discussion of the granular features of merit of the method in the early Results section (this seems more appropriate for a journal like Analytical Chemistry or another similarly focused, technical publication) but it is also probably okay as-is. I have a few comments on the reporting of the method itself, and none on the yeast signaling biology as that is outside of the scope of my expertise.

I am glad to see the authors responsibly reference the Batth et al. publication that came out in MCP just a few weeks before they submitted this manuscript - that reference only reinforces the present work as technically sound. That is very much appreciated, and in my view does not at all impact novelty. The question of novelty lies perhaps in the general idea of automation for sample handling, and is a matter of perspective - here is a method that is really helpful and powerful, even if it isn't the most conceptually novel thing I've ever heard of.

The phosphopeptide purification discussion seems a little out of alignment with some of the recent discussion in the literature regarding the selectivity of different resins - the authors are encouraged to review Ruprecht, Kuster & Lemeer (MCP 2015), cite it here and comment on it in the context of

the multiple resin purification analysis described in their data. In that study, Fe-IMAC was effectively comprehensive in its phosphopeptide isolation capacity, rather than complementary. Some discussion of that point (is it batch versus column format that generates this difference?) would be helpful for readers.

In summary, this is a strong technical report with a lot of potential for making one of the most imprecise and variable aspects of proteomics methods more tractable for the field, which is a critical step for harmonizing inter-laboratory comparisons and make data integration much easier in the future than it is now.

Reviewer #3:

Leutert et al. describe development of a phosphoproteomic pipeline, which is based on automated SP3 digestion followed by phosphopeptide enrichment. The authors then went on to apply this high throughput method to the analysis of yeast phosphoproteomics under various conditions using DIA proteomics workflow. Overall, this is a solid study, which shows the robustness of these methodologies and presents a novel way of automating a rather complex sample preparation pipeline. It is also very nicely written and presented, and the added protocol may be of value to future users of the techniques. The title and the focus of the main part of the manuscript suggest that the method development is their major emphasis. Indeed, it includes some interesting insights, such as the effects of protease and phosphatase inhibitors and the general automation principle. However, these advancements seem to me of rather limited interest to the broad systems biology community. In contrast, the application to yeast may be of interest to a broader community, if the authors decide to add some integrated analyses and emphasize this part a bit more. In addition, I add some specific comments:

1. In several cases, it seems like the text does not accurately reflect the figures. For example, in page 5 the authors indicate that the optimal conditions for R2-P1 were binding at 50% or 80% ethanol at pH 8 and washing the beads with 80% ethanol. However according to Fig. S1A it seems that ACN at pH2 gives better results. Similarly, the authors indicate that the advantage of R2-P2 is especially in the cases of low input amounts. However, in Fig. 2 (the comparison between SepPak and R2-P2) it is not convincing that there is an actual advantage of R2-P2, since with 50ug input it seems to have worse results. Also, reduced number of phosphopeptides in the 400 ug sample is a bit puzzling, and the authors do not address it. If this is a result of some technical variance, it suggests that the differences between the two methods are negligible.
2. Some of the analytical processes are not sufficiently explained, and raise some questions regarding the actual meaning of the results. For example, it is not clear how the data were filtered, and what data (filtered/imputed) was used for the statistical tests. Was this performed for each treatment separately, or did they request minimum 2/3 in all treatments? Were missing values replaced by a single constant value? The sentence in page 10 is unclear "An average of 7,880 phosphorylated peptides were quantified in at least two out of three biological replicates for each condition and 40% of the phosphopeptides had all sites confidently localized". Does it mean that the statistical experiments were performed only on these 7880 sites (on average per condition)? Why would only 40% of phosphopeptides have confident localization? I would expect much higher percentages, especially given that most peptides are singly phosphorylated. In the enrichment analyses, they should indicate what the background was.
3. Figure 5C shows enrichment of GO categories within the different clusters (from 5B). This analysis shows multiple repeated categories (as expected from GO), such as translation, endocytosis, metabolic process etc. This functional overlap between clusters reduces the value of the analysis, as it is impossible to associate between the biology of the treatment and the changes. The authors should dig deeper into the analyses and present the specificity of the processes (or sub-processes), and potential overlap between the categories. They should also refer to the time course that they performed, and show how these proteins and processes change with the timeline of the experiment. The way the data is presented in the current version, loses this dimension.
4. Following the enrichment analysis (Figure 5), Figure 6 focuses on specific signaling pathways. A systems biology study should aim to connect the specific pathways and present how the signal is transmitted downstream to activate the enriched processes.

A couple of minor points:

5. The authors should indicate the model of mass spectrometer.
6. I assume that in page 18 the authors meant $FDR < 0.01$.
7. Dataset3 should include only phosphopeptides that appear in at least two of three replicates. However, it includes approximately 500 peptides that appear in only one sample altogether. In addition (and could be related), sample `unt_10_1` is missing.

RESPONSE TO REVIEWERS

Reviewer #1:

In this manuscript, Leutert et al have implemented a robotic liquid handling procedure for sample preparation in a phosphoproteomics workflow. Specifically, they combine SP3, a previously developed method for protein clean up from cell extracts, with phospho-peptide enrichment in an integrated procedure on a KingFisher based on the use of magnetic beads in both parts of the workflow. The authors benchmark performance of the system compared to conventional methods, showing high reproducibility and scalability. They next apply the methodology to a phosphoproteomic analysis of yeast to probe response to various perturbations, using a data-independent mass spectrometric approach, and they perform pathway analysis to identify changes in phosphorylation events that are unique or shared between the treatments, denoting treatment-specific response or crosstalk. Authors conclude that the presented workflow should be broadly applicable for high-throughput applications under standardized conditions.

This is a well-designed study, addressing a general need in proteomics to enhance standardization in proteomic sample preparation to improve reproducibility and throughput. The presented procedure implemented on a KingFisher liquid handling robot is a highly relevant contribution in this direction, taking advantage of magnetic bead-based procedures for protein clean-up and phosphor-peptide enrichment by SP3 and metal affinity beads, respectively. The latter type have been extensively used in the literature in various configurations the literature, however not on this type of robotic system. SP3 is a relatively novel method for generic protein clean-up now implemented on a robot for the first time. To assess the performance of a platform as introduced here, both bead types should be evaluated separately and in combination with appropriate metrics to indicate reproducibility, variance, sensitivity etc. As detailed below, one chief concern is that the authors could have been somewhat more rigorous in scrutinizing performance. The application of the established workflow to various stresses in yeast is nice and relevant, showing that the R2-P2 can be effectively used to probe multiple samples to disclose new biology. Yet, none of this (e.g. pathway crosstalk) was further validated, however this may not be a strict requirement for a paper with a methods focus.

Specific comments:

1. Page 6: Assessing quantitative reproducibility is key to demonstrate robust performance of the presented R2P2 platform. However this was only shown for an experiment with just 3 replicates (Fig 1E), and it is unclear if these samples were prepared within one batch, or independently across batches, ideally over an extended period of time. Such data should be shown to more convincingly demonstrate stable performance with low variability.

The replicates (n=3) shown for each figure comparing different parameters of the method were always prepared in one batch; we clarify this now in the method section. To address the reviewers comment we have performed a new experiment to assess the performance and

reproducibility of R2-P1 and R2-P2 comparing replicates of the same robotic run and replicates of separate robotic runs conducted on different days. For this, we separated a yeast cell lysate in 25 aliquots and performed 5 separate runs of R2-P1 followed by R2-P2 (on 5 different days), with 5 replicates each. We included a new section called “Quantitative reproducibility of R2-P1 and R2-P2” to describe our results, which are shown in Fig 2E-H and Fig EV5. These results further support the reproducibility of the method and demonstrate that the method is capable to differentiate biological from technical differences.

2. For the SP3-based method, the number of identified phosphopeptides does not scale with the amount of input material (Fig 2A). In fact, only slightly more phosphopeptides were identified from 400ug starting material than from 25ug. Authors should clarify whether this is due to SP3 or phospho-enrichment, or rather sub-optimal interfacing of the two. Judging from expected scaling when using SepPack (left part of Fig 2A), the effect seems to be due to SP3. This begs for an explanation, or for more replicates (just 2 as shown may not suffice) to verify reproducible recovery of proteins from high-input samples by SP3, with and without subsequent phospho-enrichment. Furthermore it must be ruled out that variability is due to suboptimal sample handling on the KingFisher, which normally speaking should outcompete manual processing by SepPack in terms of reproducibility. Also, it should be excluded that variance is due to the absence of phosphatase inhibitors in the SP3-based method, while these are present (?) when using SepPack.

In the scalability experiment, the most important metric is on the quantitative/MS1 intensity scaling. Here, we observe the expected scaling of peptide abundances with increasing protein input amounts for both SepPak and R2-P2. We have expanded our analysis on the quantitative scalability of R2-P2 and provide metrics on the linear scaling of MS1 phosphopeptide intensities with protein input amounts (Fig 2B-D).

However, as the reviewer points out, the number of peptide identifications did not scale. Because peptide identifications can be limited by the speed of the MS instrument and the number of MS/MS collected (in our case gradients were fairly short: 60 min), we would not necessarily expect scaling here. However, we were surprised that for R2-P2 we observed fewer identifications with 400µg protein than with 200µg. Taking suggestion of the reviewer, we decided to investigate this result a bit further. As a clarification, the experiment was conducted with 3 replicates and not 2. And both SPE and R2-P2 were performed on the same lysate with no protease or phosphatase inhibitors.

We observed however that the phosphopeptide enrichment efficiencies for this particular experiments were rather low (around 60%) and varied across samples (Fig EV4C), and this variation encoded the differences in peptide identifications (now shown in Fig EV4B). To further investigate this, we conducted an additional R2-P2 experiment for the relevant protein input range of 100 µg - 400 µg in quadruplicates. This time phosphopeptide enrichment efficiencies were > 95% (as in all the other experiments presented in the manuscript) and we observe

minimal variation across samples. In this new experiment, phosphopeptide identifications as well as phosphopeptide MS1 intensities scaled with the input protein amount (Fig EV4D-F).

3. The authors claim that R2-P2 has higher sensitivity than conventional systems (page 8 and abstract). It is not clear which observations lead to this conclusions, since scaling experiments were tested towards higher input, not lower input.

We based this statement on the number of phosphopeptide identifications obtained with the R2-P2 method compared to the SepPak method for 25µg protein input amounts (Fig EV4B). However the reviewer is correct that we have not extensively tested the range for low protein input amounts. The focus of this study is on high-throughput and automation and we have therefore removed claims that R2-P2 is more sensitive than classical methods.

4. Page 9: Reproducible preparation of 96 samples is claimed, however data for only 3 replicates was presented. Therefore claims should be toned down, or more samples should be analysed.

We apologize for the poor wording here. We toned down statement to only imply that the method can in principle handle 96 samples in parallel. Also, the novel results for replicate analysis (5x5 replicates) we present in the revision indicate that it should be straightforward to process 96 samples in parallel and analyze samples prepared in different batches.

5. In Fig S5A it is striking and slightly worrying that replicate samples are not always as closely spaced as one would wish. The authors claim that this is due to biological variation, especially for the 90-min conditions, however this was not substantiated. Since this is primarily a methodological paper aiming to establish a KingFisher-based workflow, it would be extremely useful to experimentally demonstrate that technical and biological variability can be distinguished, and that the former is smaller than the latter. Therefore, authors should perform replicates from the same yeast isolate (e.g. one of the 90-min conditions) and demonstrate that process-replicates are more closely spaced than biological replicates in a PCA analysis.

The newly provided replicate analysis shows that technical variability for R2-P2 is as good as one would expect for highly reproducible sample preparation with label-free MS quantification (median phosphopeptide CVs ~25% and median Pearson's correlation > 0.9) (Fig 2H, Fig EV5B). We now provide an additional figure where we plot the distribution of Pearson correlation values for same robotic run *versus* robotic runs on different days for the replicate analysis (Fig EV5C).

Along the same lines, we now also show a similar plot for the signaling experiment where we compare biological replicates and different treatment conditions resolved for the different timepoints (Fig 4C). Median Pearson's correlations are > 0.9 for biological replicate comparisons at 10 and 30min, whereas there is a bigger spread for Pearson's correlations of biological replicates for the 90min time point with median Pearson's correlations above 0.85.

Pearson's are lower for comparisons of different treatments. These results show that technical variability and variability between biological replicates are lower than variability induced by the different treatments. Taken together we are confident that our method is sufficiently robust to detect biological variability.

6. It is unclear what can be gathered from the GO analysis in Fig S5B, since it includes many and diverse biological processes across the various perturbation conditions. Also it is unclear how the authors get to the conclusion that component 2 resolves temporal effects. It may be better to remove this figure.

We agree with the reviewer that our previous Fig EV5B was simplified for the number of comparisons and thus difficult to interpret. As suggested, we have removed this figure.

7. Similarly for Fig5C: Given the diversity of GO-terms recovered between treatments or, unexpectedly, between time points, anything can be concluded from Fig 5C, and therefore at the same time nothing can be distilled. Really there is no individual or unifying conclusion that can be drawn from this analysis. To a large part this is also due to the fact that many terms are very general and therefore minimally informative (transport, response to stimulus, RNA metabolic process etc). Unsurprisingly, the text associated with this figure is highly descriptive recapitulating what one can read in the figure, without clarifying discrepancies or making biological inferences. It would have been more informative to select one treatment (with time points) and perform a more in-depth data analysis, or to leave out this figure altogether.

We have replaced this figure with a new GO analysis figure (Fig 5C) that integrates treatment and time dimensions, while reducing the number of GO terms shown. We think this figure provides a much better visualization of enriched terms and facilitates interpretation.

8. Fig5B: The basis is unclear for the general conclusion that the 90-min clusters include phosphosites regulated in multiple conditions, more so than the 10 and 30-min clusters: all 3 heatmaps show regions indicating generally changing peptides (lower sections for 10 and 30 min, middle section for 90 min).

We have updated and simplified our discussion of the heat map.

Reviewer #2:

In this report by Leutert and colleagues, the authors develop and report on a semi-automated proteomic and phosphoproteomic sample processing workflow that leverages a combination of solvent-based protein aggregation on microparticles and magnetic robot-based automation to generate reproducible digests and purifications for downstream LC-MS/MS-based analysis. As the field of proteomics moves towards higher throughput and into less specialized laboratories, the need for smarter sample processing and handling increases, which frames the rationale and significance of the work as described.

This is a really well-written manuscript that describes (in almost too fine!) details of the conceptual basis for the microaggregation-based sample handling, the careful validation experiments, and the comparisons with existing technology that appear to make their "R2P1/P2" methods as good as the status quo. Given the increase in adaptation of isobaric tagging methods in quantitative proteomics and phosphoproteomics, the sample input levels as described in the paper are well aligned with the typical downstream applications for such methods (in spite of the DIA application in this manuscript). The figures accurately and clearly convey the message and the study seems well grounded. My personal opinion is that the authors could have pared down their discussion of the granular features of merit of the method in the early Results section (this seems more appropriate for a journal like Analytical Chemistry or another similarly focused, technical publication) but it is also probably okay as-is. I have a few comments on the reporting of the method itself, and none on the yeast signaling biology as that is outside of the scope of my expertise.

We thank the reviewer for the positive review. As mentioned by the reviewer we now include a sentence in the discussion that we anticipate the method to be combinable with isobaric labeling of samples.

I am glad to see the authors responsibly reference the Batth et al. publication that came out in MCP just a few weeks before they submitted this manuscript - that reference only reinforces the present work as technically sound. That is very much appreciated, and in my view does not at all impact novelty. The question of novelty lies perhaps in the general idea of automation for sample handling, and is a matter of perspective - here is a method that is really helpful and powerful, even if it isn't the most conceptually novel thing I've ever heard of.

The phosphopeptide purification discussion seems a little out of alignment with some of the recent discussion in the literature regarding the selectivity of different resins - the authors are encouraged to review Ruprecht, Kuster & Lemeer (MCP 2015), cite it here and comment on it in the context of the multiple resin purification analysis described in their data. In that study, Fe-IMAC was effectively comprehensive in its phosphopeptide isolation capacity, rather than complementary. Some discussion of that point (is it batch versus column format that generates this difference?) would be helpful for readers.

We agree with Ruprecht, Kuster & Lemeer that the apparent complementarity of the different enrichment methods is primarily caused by limited binding capacity and/or inefficient elution in the batch regime, and/or the stochasticity of DDA mass spectrometry. Similar to their results, in our hands Fe-IMAC performs the best as judged by total phosphopeptide intensity and number of identifications. We have re-evaluated and edited our claims regarding the phosphopeptides detected exclusively with one enrichment approach, mention the suggested reference in the discussion, and focus the conclusions of the section on the versatility of R2-P2 to work with a diversity of phosphopeptide enrichment methods.

In summary, this is a strong technical report with a lot of potential for making one of the most imprecise and variable aspects of proteomics methods more tractable for the field, which is a critical step for harmonizing inter-laboratory comparisons and make data integration much easier in the future than it is now.

Reviewer #3:

Leutert et al. describe development of a phosphoproteomic pipeline, which is based on automated SP3 digestion followed by phosphopeptide enrichment. The authors then went on to apply this high throughput method to the analysis of yeast phosphoproteomics under various conditions using DIA proteomics workflow. Overall, this is a solid study, which shows the robustness of these methodologies and presents a novel way of automating a rather complex sample preparation pipeline. It is also very nicely written and presented, and the added protocol may be of value to future users of the techniques. The title and the focus of the main part of the manuscript suggest that the method development is their major emphasis. Indeed, it includes some interesting insights, such as the effects of protease and phosphatase inhibitors and the general automation principle. However, these advancements seem to me of rather limited interest to the broad systems biology community. In contrast, the application to yeast may be of interest to a broader community, if the authors decide to add some integrated analyses and emphasize this part a bit more.

The reviewer is correct that the focus of this paper is a method, specifically we present a method that will enable systems biology studies of cellular signaling. We think it fits nicely the "Method Article" format of Molecular Systems Biology. However, we value the advice from the reviewer on emphasizing the application part of the manuscript a bit more to appeal to a broader audience. Accordingly, we have modified the title, and improved the analysis and presentation of the MAPK dataset.

In addition, I add some specific comments:

1. In several cases, it seems like the text does not accurately reflect the figures. For example, in page 5 the authors indicate that the optimal conditions for R2-P1 were binding at 50% or 80% ethanol at pH 8 and washing the beads with 80% ethanol. However according to Fig. S1A it seems that ACN at pH2 gives better results.

It is true that 80% ACN at pH2 and 50% and 80% EtOH at pH8 provided very similar results. Using 50% EtOH for binding provides the advantage that larger volumes of lysate can be processed. 80% ACN at pH2 makes the beads very sticky and increases the risk of inefficient bead handling on the magnetic processor due to clumping. We now describe this in the paper.

Similarly, the authors indicate that the advantage of R2-P2 is especially in the cases of low input amounts. However, in Fig. 2 (the comparison between SepPak and R2-P2) it is not convincing that there is an actual advantage of R2-P2, since with 50ug input it seems to have worse

results. Also, reduced number of phosphopeptides in the 400 ug sample is a bit puzzling, and the authors do not address it. If this is a result of some technical variance, it suggests that the differences between the two methods are negligible.

We acknowledge that we have not extensively tested low input amounts for R2-P1 and R2-P2 and therefore removed the claims that our method is more sensitive to classical methods (abstract and page 8).

Also, as explained in the answers to reviewer 1, we have investigated the reason for the reduced number of phosphopeptide identifications in the 400µg R2-P2 sample and found that the number of phosphopeptide identifications reflected differences in enrichment efficiencies in this particular experiment (Fig EV4B,C) (we note that enrichment efficiencies for all other experiments presented in the manuscript are >95%). However, for the sake of scalability and application to quantitative phosphoproteomics, the most important metric of this experiment is the linear scaling of phosphopeptide intensities, which was observed for both SPE coupled with automated IMAC and R2-P2. We provide additional figures for this (Fig 2B-D). Also, as a sanity check for the 400µg (lower than expected) phosphopeptide identification result, we have repeated R2-P2 for the 100µg, 200µg, and 400µg input protein amounts in quadruplicates. In this new experiment, R2-P2 achieved the expected enrichment efficiency of >95%, and both phosphopeptide identifications and intensities scaled up with increasing input amounts (Fig EV4D-F).

2. Some of the analytical processes are not sufficiently explained, and raise some questions regarding the actual meaning of the results. For example, it is not clear how the data were filtered, and what data (filtered/imputed) was used for the statistical tests. Was this performed for each treatment separately, or did they request minimum 2/3 in all treatments? Were missing values replaced by a single constant value?

The sentence in page 10 is unclear "An average of 7,880 phosphorylated peptides were quantified in at least two out of three biological replicates for each condition and 40% of the phosphopeptides had all sites confidently localized". Does it mean that the statistical experiments were performed only on these 7880 sites (on average per condition)?

The data for the DIA signaling experiment was filtered in the following way:

- For all biological replicates (same condition, same time point), at least 2 out of 3 values were required otherwise the phospho peptide was considered not to be present.
- In this filtered peptide set, missing values were replaced by 0.
- Phosphopeptides containing the same combination of phosphosites were aggregated into phosphopeptide isoforms by summing their intensity values.
- This resulted in an overall of 8,314 phosphopeptide isoforms.
- This data was used for statistical tests and plots.

We have revised our methods section, results, and figure legends, providing more details on the bioinformatic analysis, which help clarify the points mentioned by the reviewer.

Why would only 40% of phosphopeptides have confident localization? I would expect much higher percentages, especially given that most peptides are singly phosphorylated.

We were not surprised by this result. We note that our localization score cut off is stringent ($p < 0.01$) (compared to commonly used confidence of 75% in MaxQuant) and that our data set contains a high percentage of multiple phosphorylated peptides. If one out of two or three phosphorylation sites in a peptide cannot be confidently localized the peptide was classified as not having a confident localization.

In the enrichment analyses, they should indicate what the background was.

We now state more clearly what the background of our GO enrichment analysis was. For the analysis of regulated sites over untreated the whole yeast proteome was taken as a background (new Fig 5C).

3. Figure 5C shows enrichment of GO categories within the different clusters (from 5B). This analysis shows multiple repeated categories (as expected from GO), such as translation, endocytosis, metabolic process etc. This functional overlap between clusters reduces the value of the analysis, as it is impossible to associate between the biology of the treatment and the changes. The authors should dig deeper into the analyses and present the specificity of the processes (or sub-processes), and potential overlap between the categories. They should also refer to the time course that they performed, and show how these proteins and processes change with the timeline of the experiment. The way the data is presented in the current version, loses this dimension.

We agree with the reviewer that Figure 5C was too general. We have therefore replaced Figure 5C with a new GO analysis that integrates the time dimension and the different treatments. We believe that this figure more clearly outlines the differences and similarities between the different treatments and shows how the enrichment of phosphorylation events for different GO terms changes over treatment time.

4. Following the enrichment analysis (Figure 5), Figure 6 focuses on specific signaling pathways. A systems biology study should aim to connect the specific pathways and present how the signal is transmitted downstream to activate the enriched processes.

While some connections can be made between specific pathways, kinases, and terminal points of the pathway (enriched processes), mostly based on prior literature; making all these connections in a systematic, data-driven way would require a larger dataset including a greater number of conditions and timepoints. The novelty of this manuscript lays in the method and we briefly showcase how the method can be applied to systems biology questions. While we agree this will be very interesting, we consider this expanded analysis beyond the scope of the current paper.

A couple of minor points:

5. The authors should indicate the model of mass spectrometer.

We indicate the model of mass spectrometers we used in the method section.

6. I assume that in page 18 the authors meant $FDR < 0.01$.

Indeed we meant $FDR < 0.01$, we have corrected this mistake.

7. Dataset3 should include only phosphopeptides that appear in at least two of three replicates. However, it includes approximately 500 peptides that appear in only one sample altogether. In addition (and could be related), sample unt_10_1 is missing.

Thanks for spotting this mistake. Indeed there were untreated samples that were not used for the analysis in this table. We have removed these samples from the table and the peptides now appearing in the table have been measured at least twice in individual biological replicates.

Thank you for sending us your revised manuscript. We have now heard back from the three referees who agreed to evaluate your study. As you will see below, the reviewers are now satisfied with the modifications made and think that the study is suitable for publication, pending rather minor text modifications requested by reviewer #1.

REFEREE REPORT

Reviewer #1:

The authors have appropriately addressed my concerns, greatly aided by the addition of new data. One minor question remains with regard to point 2: In the new data that are presented in Fig EV4D-F, it is comforting to see that output now scales with input. However, it is still unclear what (if anything) was changed in the R2-P2 procedure or set-up to achieve this result, compared to what was shown in Fig 2A in the first submission. Commenting on this in the manuscript will be helpful for prospective users of the method, pointing them to potential critical aspects of the procedure.

Reviewer #2:

The authors have addressed my comments.

Reviewer #3:

The authors answered all my questions, expanded the necessary sections and corrected a couple of minor mistakes. I therefore find the manuscript suitable for publication.

Corresponding Author Name: Judit Villen, Mario Leutert

Manuscript Number: MSB-19-9021